# An anatomical and physiological basis for flexible coincidence detection in the auditory system

**Lauren J Kreeger[1], Suraj Honnuraiah[1,2], Sydney Maeker[1], Siobhan Shea[1], Gordon Fishell[1,2], Lisa Goodrich[1]***

[1]Harvard Medical School, Department of Neurobiology, Boston, United States; [2]Stanley Center for Psychiatric Research, Broad Institute of MIT and Harvard, Cambridge, United States

**\*For correspondence:**
Lisa_Goodrich@hms.harvard.edu

**Competing interest:** The authors declare that no competing interests exist.

## eLife Assessment

Kreeger et al. **convincingly** demonstrate that octopus cells in the mouse cochlear nucleus, previously thought to rely primarily on excitatory inputs for coincidence detection, also receive glycinergic inhibitory synaptic inputs that influence their synaptic integration. Using advanced techniques, including genetic mouse models, optogenetics, microscopy, slice physiology, and computational modeling, this **important** study reveals that inhibition can shunt synaptic currents and alter the timing of dendritic EPSPs, both of which are significant for auditory processing. This research broadens the understanding of octopus cells' roles in sensory processing, highlighting the importance of inhibitory inputs in shaping fast, high-frequency neural response capabilities.

**Abstract** Animals navigate the auditory world by recognizing complex sounds, from the rustle of a predator to the call of a potential mate. This ability depends in part on the octopus cells of the auditory brainstem, which respond to multiple frequencies that change over time, as occurs in natural stimuli. Unlike the average neuron, which integrates inputs over time on the order of tens of milliseconds, octopus cells must detect momentary coincidence of excitatory inputs from the cochlea during an ongoing sound on both the millisecond and submillisecond time scale. Here, we show that octopus cells receive inhibitory inputs on their dendrites that enhance opportunities for coincidence detection in the cell body, thereby allowing for responses both to rapid onsets at the beginning of a sound and to frequency modulations during the sound. This mechanism is crucial for the fundamental process of integrating the synchronized frequencies of natural auditory signals over time.

## Introduction

Perception depends on the ability of neurons to encode discrete features of the sensory environment. To generate accurate percepts of complex auditory stimuli, neurons must compute both what frequencies are present and when those frequencies occur across multiple time scales. For instance, overlapping sounds, such as two competing speakers in a noisy room, are distinguished perceptually by correctly binding frequencies with coherent onsets and synchronized temporal modulations (*Shamma et al., 2011*; *Pressnitzer et al., 2011*). Such computations require coincidence detection that can encode co-occurring frequencies with submillisecond precision. Frequency information is communicated by spiral ganglion neurons (SGNs), whose central axons, also called auditory nerve fibers, project through the eighth nerve, bifurcate, and spread tonotopically to fill each division of

**eLife digest** Imagine trying to listen to a friend in a busy coffee shop. Your brain helps to focus on their voice by distinguishing between the different sounds in the environment. This ability relies on specialized neurons in the auditory system called octopus cells, which detect when sound frequencies occur and change together.

Unlike most neurons in the auditory system, octopus cells can be activated by inputs from many different frequencies. However, they only reliably release an electrical signal after receiving these excitatory inputs simultaneously. This explains why these cells can respond at the beginning of a sound stimulus but not why they can also react to frequencies that change together over time, like in speech or music. This suggests that the cells' computations may be more complex than previously thought. Rather than relying solely on excitatory inputs, alternative signals that reduce activity, known as inhibitory inputs, may also play a role.

To test this hypothesis, Kreeger et al. studied genetically modified mice to reveal the octopus cells' activity using molecular fluorescent labels. They created a map of the incoming signals to the octopus cells, showing inhibitory inputs to the outer branched extensions of the cells. By using light to control cell activity and measuring electrical responses, Kreeger et al. showed that inhibition helps refine how excitatory signals travel through the neuron. This gives the octopus cell more time to process excitatory inputs, meaning that one cell can make computations both about the onset of sound and about frequencies that occur together during the sound.

These results provide a fuller picture of how the brain helps distinguish between sounds in noisy environments. Understanding the fine-tuning role of inhibitory signals in octopus cells may help researchers improve hearing aids and develop treatments for auditory processing disorders.

the cochlear nucleus complex (CNC; *De No, 1933*; *Fekete et al., 1984*; *Liberman, 1991*; *Brown and Ledwith, 1990*; *Leake and Snyder, 1989*; *Figure 1A*). In addition, SGNs fall into physiologically distinct subtypes that are recruited at different intensities, allowing sounds to be detected across a wide dynamic range (*Sachs and Abbas, 1974*; *Liberman, 1982*; *Liberman, 1978*; *Evans and Palmer, 1980*; *Taberner and Liberman, 2005*; *Palmer and Evans, 1980*; *Winter et al., 1990*). Many target neurons receive SGN inputs from a limited range of frequencies, encoding only a fraction of the frequency components within a stimulus. This presents a challenge for auditory circuits which must ultimately bind co-occurring frequencies while retaining sequence information in order to locate and recognize sounds. Understanding how these first computations are made is a key step towards deciphering the basis of perception.

In the mammalian auditory system, precise encoding of broadband timing information begins with the octopus cells of the CNC. Octopus cells are excitatory neurons that bind together co-occurring frequency information on a submillisecond timescale and send this information along one of the parallel ascending pathways in the auditory brainstem. Octopus cells are named for their large-diameter tentacle-like dendrites (*Osen, 1969a*; *Osen, 1969b*), which are oriented unidirectionally across a tonotopic array of SGNs such that each neuron integrates inputs from a wide range of frequencies (*Godfrey et al., 1975*; *Recio-Spinoso and Rhode, 2020*; *Rhode et al., 1983*; *Ritz and Brownell, 1982*). SGNs provide the major excitatory inputs onto octopus cells. Biophysically, octopus cells have low input resistances near rest (~4 MΩ; *Golding et al., 1999*; *Bal and Oertel, 2000*), fast time constants (~200 μs; *Golding et al., 1999*; *Bal and Oertel, 2000*; *Bal and Oertel, 2001*), and large low-voltage-activated potassium (~40 nS at rest; *Golding et al., 1999*; *Bal and Oertel, 2001*), and hyperpolarization activated (~62 nS at rest; *Golding et al., 1999*; *Bal and Oertel, 2000*) conductances. Together these properties give octopus cells impressively narrow windows of coincidence detection on the order of 1 ms (*Cao and Oertel, 2017*; *Golding et al., 1995*; *Golding and Oertel, 2012*; *Manis and Marx, 1991*; *Oertel, 1983*). This combination of receiving SGN innervation from broad frequencies and their biophysical specializations establish octopus cells as spectrotemporal coincidence detectors that can reliably encode the timing of complex stimuli, such as the broadband transients found in speech and other natural sounds (*Golding et al., 1999*; *McGinley, 2014*; *McGinley et al., 2012*). Fittingly, in vivo recordings from octopus cells demonstrate their ability to phase lock to broadband transients at rates up to 1 kHz (*Lu et al., 2018*; *Smith et al., 2005*; *Smith*

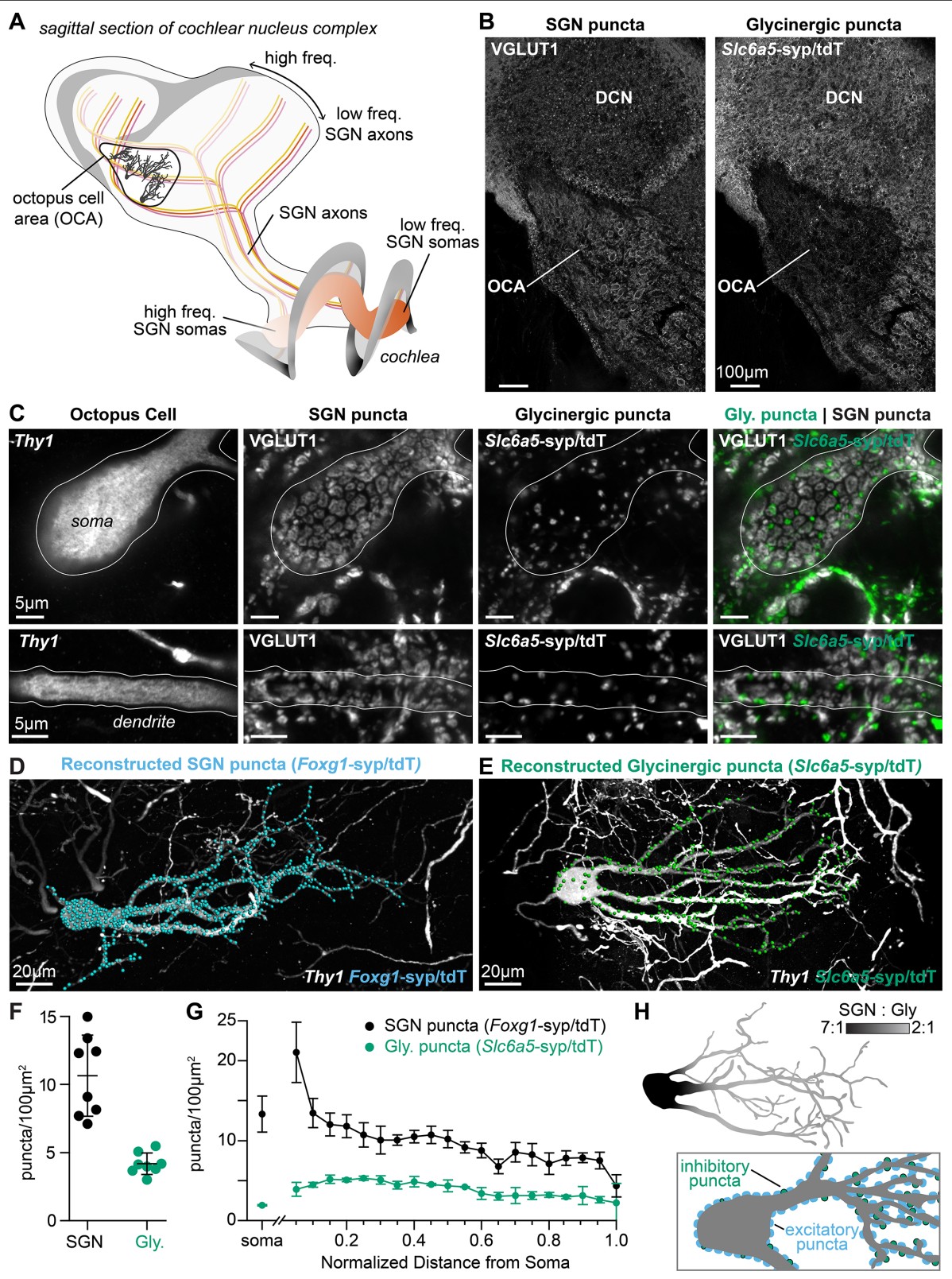

**Figure 1.** Excitatory and inhibitory synapses to octopus cells form two domains. (**A**) Illustration of spiral ganglion neuron (SGN) central axons branching within a parasagittal section of the mouse cochlear nucleus complex (CNC). SGN somas in the cochlea are tonotopically organized according to frequency. Axons remain organized throughout the ventral (VCN) and dorsal (DCN) divisions of the CNC. Octopus cells are found in the octopus cell area (OCA) of the VCN. (**B**) Excitatory SGN puncta labeled with a VGLUT1 antibody (left) and glycinergic puncta labeled with *Slc6a5*Cre-dependent syp/

*Figure 1 continued on next page*

*Figure 1 continued*

tdT (*Slc6a5*-syp/tdT; right) in a parasagittal section of the CNC. The teardrop-shaped OCA is not devoid of inhibitory inputs, although they are less prominent than in the surrounding CNC. (**C**) A *Thy1* sparsely labeled octopus cell with excitatory SGN (VGLUT1) and inhibitory (*Slc6a5*-syp/tdT) puncta. Micrographs of 3 μm confocal z-stacks show puncta on the medial surface of a soma (top) and a dendrite (bottom). (**D**) Representative reconstruction of excitatory SGN puncta labeled with *Foxg1*^Cre-dependent syp/tdT (*Foxg1*-syp/tdT; blue) onto a *Thy1* sparsely labeled octopus cell (white). (**E**) Representative reconstruction of inhibitory puncta labeled with *Slc6a5*-syp/tdT (green) onto a *Thy1* sparsely labeled octopus cell (white). (**F**) Puncta density for excitatory SGN (*Foxg1*-syp/tdT; black: 10.7 ± 3.0, n=8 cells, 4 mice) and glycinergic puncta (*Slc6a5*-syp/tdT; green: 4.2 ± 0.8, n=8 cells, 3 mice) on octopus cell dendrites. Data are presented as mean ± SD. (**G**) Puncta density on somas for excitatory SGN (black: 13.3 ± 2.2, n=8 cells, 4 mice) and glycinergic puncta (green: 1.8 ± 0.1, n=8 cells, 3 mice) and the density along the length of dendrites. Data are presented as mean ± SEM. (**H**) *Top:* Illustration of an octopus cell and the ratio between excitatory SGN puncta and glycinergic puncta. *Inset:* Illustration of an octopus cell and the relative innervation densities of excitatory SGNs (blue) and inhibitory puncta (green).

The online version of this article includes the following source data for figure 1:

**Source data 1.** Data included in *Figure 1F*.

**Source data 2.** Data included in *Figure 1G*.

*et al., 1993*). Moreover, computational models of octopus cells demonstrate that onset responses are governed by the cell's biophysical specializations and are, in large part, the result of temporal summation of excitation (*Cai et al., 2000*; *Cai et al., 1997*; *Kipke and Levy, 1997*; *Levy and Kipke, 1998*; *Levy and Kipke, 1997*; *Rebhan and Leibold, 2021*; *Spencer et al., 2012*). The simplicity of its connectivity combined with the precision of its temporal computations makes the octopus cell an attractive model for understanding how specialized anatomical and electrophysiological properties contribute to neuronal computations.

Although the octopus cell's integration of SGN inputs within a narrow time frame enables exceptional coincidence detection, such a model does not explain how other temporal features of sound stimuli are encoded. Indeed, octopus cells encode spectrotemporal sequences within their broadly-tuned response areas, like frequency modulated sounds, that likely require further circuit specializations (*Lu et al., 2022*). Although somatic depolarization can be sufficient to activate an octopus cell (*Cai et al., 1997*), the vast majority of synapses are found on dendrites. Further, SGN inputs are organized tonotopically along octopus cell dendrites, with inputs from high-frequency regions located more distal than those from low frequency regions. Dendritic morphology, passive cable properties, active resting membrane properties, and the spatial relationship between synaptic inputs can all impact excitatory post synaptic potential (EPSP) summation as excitation sweeps across the dendritic arbor and towards the soma. This raises the possibility that computations made in the dendrites influence the effective window of coincidence detection by the octopus cell. Such mechanisms could enable flexible processing that is adaptive to the dynamics of the environment without compromising high-fidelity coincidence computations.

Here, we sought to define the circuit mechanisms that allow octopus cells to act as coincidence detectors across time scales. We generated a comprehensive anatomical and physiological map of excitatory and inhibitory synaptic inputs onto octopus cell somas and dendrites and examined how this circuit organization influences octopus cell activation. Through a combination of in vitro experiments and computational modeling, we show that somatic summation of excitation is shaped by dendritic inhibition. Thus, octopus cells depend on compartmentalized computations that enable preservation of timing information both at the moment of stimulus onset and within an extended window for evidence accumulation, which is fundamental for the spectrotemporal integration of natural auditory stimuli.

## Results

### The balance of excitatory and inhibitory synapses is different in somatic and dendritic domains

To determine the wiring pattern that drives octopus cell computations, we generated a detailed map of excitatory and inhibitory synaptic inputs (*Figure 1*). Overall, octopus cells receive abundant excitatory VGLUT1+ innervation from SGNs (*Gómez-Nieto and Rubio, 2009*; *Zhou et al., 2007*) and sparse inhibitory innervation from glycinergic neurons, as visualized using the glycinergic Cre driver *Slc6a5*^Cre and the Ai34 synaptophysin-tdTomato (syp/tdT) fusion protein reporter (*Figure 1B*). The

*Slc6a5*-syp/tdT inhibitory inputs nestle between SGN inputs, especially on octopus cell dendrites (*Figure 1C*).

Quantification of the number and distribution of presynaptic puncta onto octopus cells revealed marked differences in the ratio of excitation and inhibition in the somatic and dendritic compartments. Since innervation patterns have never been systematically analyzed, we made three-dimensional reconstructions of 16 octopus cells and their excitatory SGN (n=8 cells, 4 mice) and inhibitory (n=8 cells, 3 mice) inputs. Octopus cells were visualized using a *Thy1* reporter and presynaptic puncta were labeled with the syp/tdT reporter driven either by *Foxg1*<sup>Cre</sup> (*Figure 1D*) or *Slc6a5*<sup>Cre</sup> (*Figure 1E*). Consistent with qualitative assessment, the density of SGN inputs was higher (10.7 ± 3.0 SGN puncta/100 μm$^2$) than that of inhibitory inputs (4.2 ± 0.8 puncta/100 μm$^2$, *Figure 1F*). Moreover, the relative proportions of excitatory and inhibitory inputs differed in the soma and dendrites (*Figure 1G*). On somas, SGNs provided dense innervation that continued on the proximal dendrite, then gradually declined with distance from the soma. By contrast, somas received very few inhibitory inputs. On dendrites, inhibitory puncta were evenly distributed. As a result, octopus cells have a strikingly different average ratio of excitatory and inhibitory puncta on the soma (7:1) and on the dendrite (5:2), suggesting that each compartment contributes differentially to the final computation made by the octopus cell (*Figure 1H*).

## The majority of excitatory synapses on octopus cells are from type Ia SGNs

Although uniformly glutamatergic, SGNs exhibit stereotyped physiological differences in response thresholds that could affect the nature of their inputs onto octopus cells and hence coincidence detection (*Liberman, 1991*; *Liberman, 1993*; *Rouiller et al., 1986*; *Rouiller and Ryugo, 1984*). There are three molecularly distinct SGN subtypes, referred to as Ia, Ib, and Ic SGNs, which correlate with previously shown physiological groups (*Liberman, 1978*; *Liberman et al., 2011*; *Liberman, 1982*; *Petitpré et al., 2018*; *Shrestha et al., 2018*; *Siebald et al., 2023*; *Sun et al., 2018*; *Figure 2A*). Therefore, we further categorized excitatory inputs based on SGN subtype identity. These can be identified with the presence of *Ntng1*<sup>Cre</sup>-dependent reporter expression (*Bolding et al., 2020*) in Ib and Ic SGNs (Ib/c) and its absence in Ia SGNs (*Petitpré et al., 2018*; *Shrestha et al., 2018*; *Sun et al., 2018*), coupled with very low to undetectable levels of calretinin (CR-) in Ic SGNs, and moderate to high levels of calretinin (CR+, CR++) in Ib and Ia SGNs (*Figure 2B*). As determined using the Ai14 tdTomato (tdT) reporter, *Ntng1*<sup>Cre</sup>-labeled Ib/c SGNs accounted for 60.1 ± 2.6% of the entire population, with 28.5 ± 12.2% Ib SGNs, 31.6% Ic SGNs, and 39.9 ± 2.6% Ia SGNs (*Figure 2C*: n=1599 neurons, 4 mice; mean ± SD). These proportions matched scRNA-seq estimates (*Figure 2C*, dotted lines), indicating that this approach provides full coverage. SGN subtype identity was further confirmed by examining the spatial organization of SGN peripheral processes en route to the inner hair cells (IHCs) in the cochlea, with Ia processes positioned deeper than Ib and Ic processes (*Figure 2A*, *Figure 2—figure supplement 1A-C*).

Within the ventral cochlear nucleus (VCN), where octopus cells reside, *Ntng1*<sup>Cre</sup> labeling was restricted to SGN central axons (*Figure 2—figure supplement 1D–E*). There is sparse labeling in the deep layer of the dorsal cochlear nucleus (DCN) and strong labelling throughout the thalamus, hippocampus, and cortex (*Figure 2—figure supplement 1D–E*). It is unlikely that the *Ntng1*<sup>Cre</sup>-labeled cells outside of the cochlea make synapses with octopus cells. When *Ntng1*<sup>Cre</sup>-driven tdT is co-expressed with *Foxg1*<sup>Flp</sup>-driven EYFP, all tdT-labeled axons in the VCN also expressed EYFP, suggesting that all our *Ntng1*<sup>Cre</sup> labeled inputs are a part of the *Foxg1*<sup>Flp</sup> labeled population, which are very likely to be only from SGNs in the cochlea. Additionally, in the octopus cell area, CR+ SGN central axons were segregated from *Ntng1*-tdT labeled central axons, consistent with the moderate to undetectable levels of CR in *Ntng1*-tdT SGN somas in the periphery (*Figure 2—figure supplement 1F*). Thus, *Ntng1*<sup>Cre</sup>-driven expression of syp/tdT is an appropriate tool for mapping subtype-specific connectivity onto octopus cells.

Reconstruction of *Ntng1*<sup>Cre</sup>-labeled Ib/c puncta (*Figure 2D*) demonstrated that octopus cells are dominated by inputs from Ia SGN fibers, which have the lowest response thresholds and highest rates of spontaneous activity. Octopus cell dendrites received 4.1 ± 1.0 puncta/100 μm$^2$ from Ib/c SGNs (*Figure 2E*, magenta: n=9 cells, 5 mice; mean ± SD), accounting for 38% of the total SGN density. Given that *Ntng1*-tdT+ cells account for 60.1% of the SGN population (*Figure 2C*, magenta), Ib/c

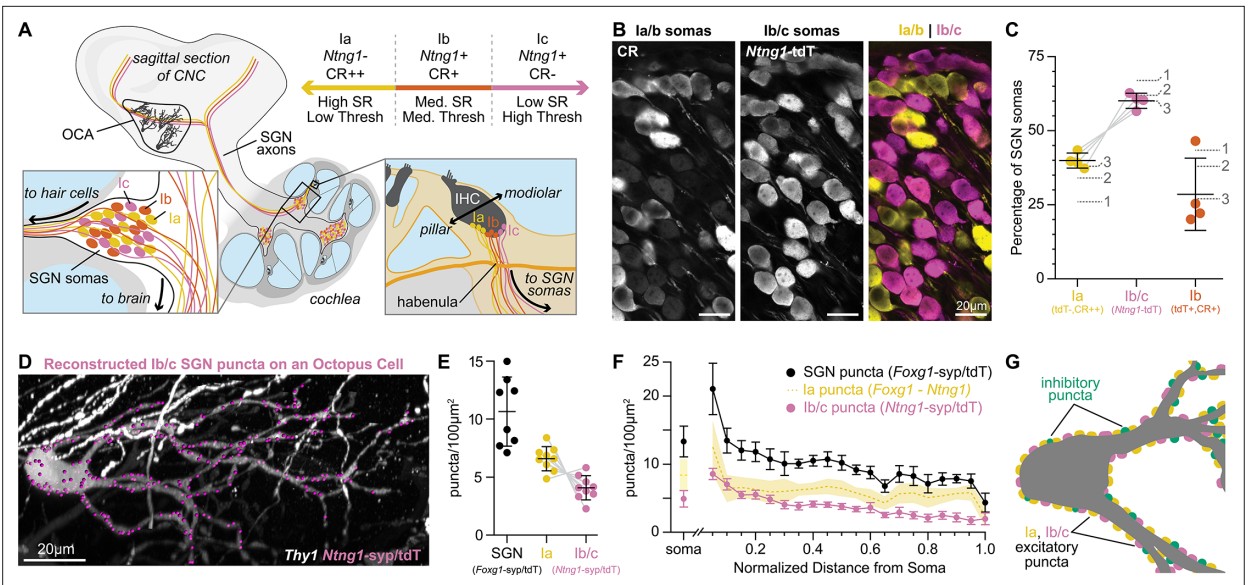

**Figure 2.** Type Ia SGNs are the primary contributors of excitation to octopus cells. (**A**) Ia (yellow), Ib (orange), and Ic (magenta) spiral ganglion neuron (SGN) axons innervate the cochlear nucleus complex (CNC). SGN peripheral processes with different properties are grossly organized along the pillar-modiolar axis as they travel through the habenula and terminate on inner hair cells (IHCs). *Ntng1*-expressing Ib/c fibers are positioned on the modiolar side (closest to the ganglion). Strongly calretinin immunopositive Ia fibers are on the other side (closest to the pillar cells). This organization correlates with spontaneous rates (SR) and thresholds (thresh) measured in vivo. Somas of all SGN subtypes are found at all tonotopic locations. (**B**) Calretinin (CR) immunolabeling distinguishes SGN subtypes. Ia/b somas label with high (CR++) and medium (CR+) levels of CR, respectively. Ic somas label with low to undetectable levels of CR (CR-). *Ntng1*[Cre]-mediated expression of tdT (*Ntng1*-tdT) labels Ib/c SGNs. (**C**) Ia SGNs (tdT-, CR++) make up 39.9 ± 2.6% of the SGN population. Ib/c SGNs (tdT+) make up 60.1 ± 2.6% of the SGN population. Ib SGNs (tdT+CR+) make up 28.5 ± 12.2% of the SGN population (n=1599 neurons, 4 mice). Data are presented as mean ± SD; individual points represent percent coverage per animal, lines connect measurements from the same animal. Dotted lines indicate percentages from 1: *Petitpré et al., 2018*, 2: *Shrestha et al., 2018* 3: *Sun et al., 2018*. (**D**) Representative reconstruction of Ib/c puncta labeled with *Ntng1*[Cre]-dependent syp/tdT (*Ntng1*-syp/tdT; magenta) onto a *Thy1* sparsely labeled octopus cell (white). (**E**) Puncta density for all SGNs (*Foxg1*-syp/tdT; black: data from *Figure 1F*), Ia SGNs (*Foxg1* - *Ntng1*; yellow: 6.6 ± 1.0), and Ib/c SGNs (*Ntng1*-syp/tdT; magenta: 4.1 ± 1.0, n=9 cells, 5 mice) along the total dendritic length. Ia density was calculated by subtracting Ib/c density from total SGN density; lines connect measurements from the same reconstruction. Data are presented as mean ± SD. (**F**) Puncta density on somas for all SGNs (*Foxg1*-syp/tdT; black: data from *Figure 1E*), Ia SGNs (*Foxg1* - *Ntng1*; yellow: 8.4 ± 2.3), and Ib/c SNGs (*Ntng1*-syp/tdT; magenta: 4.9 ± 1.2, n=8 cells, 4 mice) and the density along the length of dendrites. Data are presented as mean ± SEM. (**G**) Illustration of an octopus cell and the relative innervation densities of Ia SGNs (yellow), Ib/c SGNs (magenta), and inhibitory puncta (green).

The online version of this article includes the following source data and figure supplement(s) for figure 2:

**Source data 1.** Data included in *Figure 2C*.

**Source data 2.** Data included in *Figure 2E*.

**Source data 3.** Data included in *Figure 2F*.

**Figure supplement 1.** *Ntng1*[Cre] has high specificity for Ib/c SGNs.

**Figure supplement 2.** *Myo15*[iCre] sparsely labels Ic SGNs.

**Figure supplement 3.** Dendritic and synaptic reconstructions of octopus cells.

inputs were underrepresented on octopus cells. Octopus cells receive similarly low innervation from Ic inputs (*Figure 2—figure supplement 2G-I*: n=6 cells, 2 mice), as estimated from the degree of sparse labeling of Ic axons achieved by *Myo15*[iCre]-driven reporter expression (*Figure 2—figure supplement 2A–F*, 4.7% of SGNs) and the expected proportion of Ic SGNs in the ganglion (*Figure 2—figure supplement 2F*, dotted lines). By contrast, Ia SGNs, which comprise ~40% of the total population (*Figure 2C*, yellow), accounted for 62 ± 9.7% of SGN synapses on octopus cells (*Figure 2E*, yellow: 6.6 ± 1.0 puncta/100 μm$^2$). All three subtypes showed the same overall distribution from the soma to the distal dendrite (*Figure 2F*). Together, excitatory and inhibitory puncta densities in the innervation maps indicate the average octopus cell receives ~1035 SGN synapses (642 Ia SGN, 393 Ib/c SGN) and ~354 inhibitory synapses. Additionally, 83% of the synapses were on the dendrites, suggesting a critical role in the octopus cell computation.

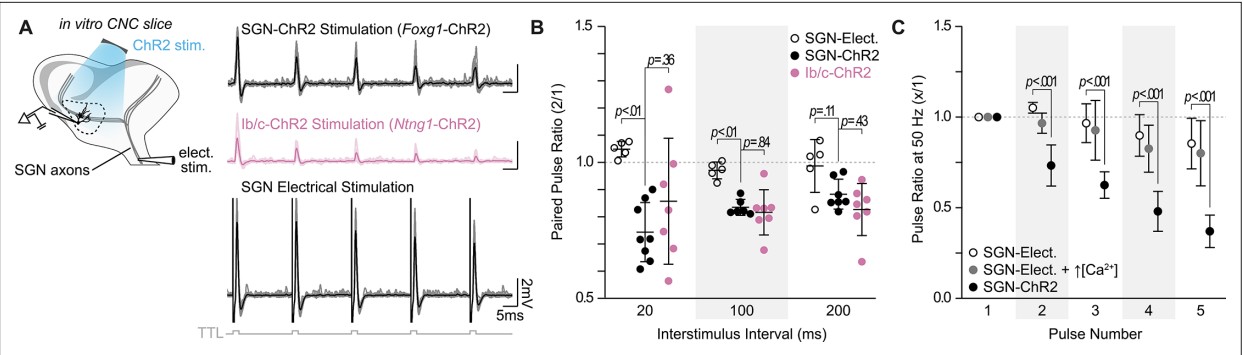

**Figure 3.** SGN subtype inputs to octopus cells do not differ in short-term plasticity. (**A**) Illustration of the experimental paradigm and representative EPSP traces recorded during in vitro whole-cell current-clamp recordings of octopus cells. Spiral ganglion neuron (SGN) central axon stimulation methods included electrical stimulation or full-field, light-evoked activation of *Foxg1*-ChR2 or *Ntng1*-ChR2 SGNs. TTL trigger pulses are shown in gray. (**B**) Paired pulse ratios for electrically stimulated SGNs (open circles: n=5 cells, 3 mice), ChR2 stimulated SGNs (*Foxg1*-ChR2; black: n=8 cells, 5 mice), and ChR2 stimulated Ib/c SGNs (*Ntng1*-ChR2; magenta: n=7 cells, 6 mice) at three interstimulus intervals. With electrical stimulation, SGN inputs to octopus cells were stable and exhibited slight facilitation at 50 Hz (20ms interstimulus interval). ChR2 stimulation caused paired pulse depression not seen with electrical stimulation. Data are presented as mean ± SD. Each data point represents the average paired pulse ratio for a cell. p values from ANOVA and subsequent Tukey HSD test are reported for comparisons between methods of SGN activation (electrical and ChR2) and SGN subpopulation composition within method of activation (SGN-ChR2 and Ib/c-ChR2). Welch's ANOVA was used for comparisons at 20ms interstimulus interval (50 Hz) as data in this condition did not meet the homogeneity of variance assumption. (**C**) Pulse ratios at 50 Hz for electrically stimulated SGNs with physiological 1.4 mM $Ca^{2+}$ ACSF (open circles: n=5 cells, 3 mice), electrically stimulated SGNs with 2.4 mM $Ca^{2+}$ ACSF (grey: n=3 cells, 2 mice) and ChR2 stimulated SGNs with physiological ACSF (*Foxg1*-ChR2; black: n=8 cells, 5 mice). Data are presented as mean ± SD. p<0.001 from ANOVA and subsequent Tukey HSD test for all comparisons between methods of SGN activation (SGN-Elect. and SGN-ChR2). There were no statistically significant differences for all comparisons between 1.4 mM (SGN-Elect.) and 2.4 mM $Ca^{2+}$ (SGN-Elect. + ↑[$Ca^{2+}$]) (p>0.100, ANOVA).

The online version of this article includes the following source data for figure 3:

**Source data 1.** Data and statistical analysis included in *Figure 3B*.

**Source data 2.** Data and statistical analysis included in *Figure 3C*.

Octopus cell reconstructions showed the same basic wiring patterns regardless of where each cell was positioned in the octopus cell area. The tonotopic position of all reconstructed octopus cell somas was estimated in 3D reconstructions aligned to a normalized CNC model of tonotopy. Octopus cells had similar morphologies (*Figure 2—figure supplement 3E–G*) and patterns of synaptic innervation (*Figure 2—figure supplement 3H–M*) regardless of where they were positioned along the tonotopic axis. Thus, Ia SGN synapses are the primary inputs to both the soma and dendrites of octopus cells. Additionally, the whole-neuron wiring diagram identifies a dendritic domain where inhibitory synapses are approximately equal in number to the Ib/c excitatory synapses from the periphery.

## SGN inputs to octopus cells facilitate at high stimulation frequencies

Whether or not an octopus cell responds to its inputs depends on when and how EPSPs travel to and then summate in the soma. To determine if SGN subtypes transmit information differently to their central targets, we performed in vitro whole-cell current clamp recordings of octopus cells (*Figure 3A*) while using Cre-dependent Channelrhodopsin-2 (ChR2) to evoke EPSPs from all SGNs (*Foxg1*-ChR2) or only Ib/c SGNs (*Ntng1*-ChR2). Octopus cells have short action potentials (~5–15 mV) that resemble their large, well-timed EPSPs (*Golding et al., 1995*; *Oertel et al., 1990*). Therefore, we evoked small EPSPs that were below spike threshold and distinguishable from action potentials using a phase plot analysis. Trains of ChR2-evoked EPSPs, ranging from 5 to 50 Hz, in both the total SGN population (*Figure 3B*, black: n=8 cells, 5 mice) and the Ib/c SGN population (*Figure 3B*, magenta: n=7 cells, 6 mice) exhibited no differences in paired-pulse plasticity at any frequency of stimulation (p>0.35 at all interstimulus intervals, Tukey's HSD), although the Ib/c population exhibited higher variability than the total SGN population (at 20ms: total SGN SD=0.11, Ib/c SGN SD=0.24).

ChR2-evoked synaptic responses are known to undergo synaptic depression (*Jackman et al., 2014*; *Zhang and Oertner, 2007*). To determine if the paired-pulse depression measured in ChR2-stimulated experiments was physiological, we used electrical stimulation to evoke EPSPs from SGNs. Electrically-evoked EPSPs had higher paired-pulse ratios than ChR2-evoked EPSPs and were mildly

facilitating at short (20 ms) intervals (*Figure 3B–C*, open circles: n=5 cells, 3 mice), consistent with an octopus cell's ability to respond reliably to click trains in vivo (*Godfrey et al., 1975*; *Rhode et al., 1983*; *Smith et al., 2005*; *Oertel et al., 2000*). In contrast, previous results using electrical stimulation demonstrated short-term depression of SGN inputs to octopus cells (*Cao et al., 2008*; *Cao and Oertel, 2010*). However, these experiments were carried out in the presence of higher, non-physiological levels of extracellular calcium. We repeated paired-pulse plasticity experiments with non-physiological calcium concentrations (2.4 mM) and similarly found that electrically evoked EPSPs from SGNs resulted in short-term depression at 50 Hz of electrical stimulation (*Figure 3C*, grey: n=3 cells, 2 mice), although not to the degree observed when using full-field, ChR2-evoked inputs (*Figure 3C*, black: n=8 cells, 5 mice).

## Glycine evokes inhibitory post synaptic potentials that are occluded by a low input resistance

Given the high density of inhibitory synapses on octopus cell dendrites, we considered the possibility that somatic and dendritic compartments contribute differently to the temporal computation made by octopus cells. A role for inhibition has not been incorporated into octopus cell models as previous efforts failed to reveal physiological evidence of functional inhibitory synapses either in vitro (*Golding et al., 1995*; *Bal et al., 2009*; *Oertel et al., 1990*) or in vivo (*Lu et al., 2022*). Similarly, we did not observe light-evoked (*Slc6a5*-ChR2) inhibitory post synaptic potentials (IPSPs) in octopus cell somas during whole-cell current clamp recordings from P30-45 mice. Since inhibitory synapses are located primarily on octopus cell dendrites, we posited that their voltage spread to the soma is limited given the extremely low input resistance of octopus cells. To decrease electrotonic isolation of the dendrites and increase input resistance, we pharmacologically reduced conductances from voltage-gated potassium (Kv) and hyperpolarization-activated cyclic nucleotide-gated (HCN) channels using 100 µM 4-Aminopyridine (4-AP) and 50 µM ZD 7288 (ZD), respectively. This cocktail increased octopus cell membrane resistance and hyperpolarized the resting membrane potential by ~8–10 mV (*Figure 4A*). To compensate for this change, membrane potentials were adjusted to within 3 mV of the original resting membrane potential with a holding current. To isolate inhibition, AMPA receptor activation was blocked using 15 µM 2,3-dioxo-6-nitro-7-sulfamoyl-benzo[f]quinoxaline (NBQX). Consistent with our hypothesis, the increase in input resistance unveiled light-evoked IPSPs in recordings from octopus cell somas (*Figure 4B*). Additionally, these IPSPs were fully abolished by bath application of 500 nM strychnine (STN) (*Figure 4C*), confirming the presence of functional glycinergic inhibitory synaptic transmission onto octopus cells.

To determine the types of glycinergic receptors contributing to IPSPs, we pharmacologically blocked subsets of glycine receptors (*Figure 4C*). IPSPs were reduced upon addition of 20 µM picrotoxin (PTX), which blocks homomeric glycine receptors (*Pribilla et al., 1992*; *Wang et al., 2006*; *Lynch et al., 1995*). Sequential addition of 100 µM cyclothiazide (CTZ), which blocks α2-containing homomeric and heteromeric glycine receptors (*Zhang et al., 2008*; *Hruskova et al., 2012*), nearly abolished the remaining IPSPs, and responses were fully abolished with further application of 500 nM STN. These results indicate that relevant glycine receptors include both large conductance extrasynaptic β-subunit lacking homomeric receptors and synaptically localized α2β receptors with slower kinetics (*Lynch, 2009*; *Veruki et al., 2007*; *Dutertre et al., 2012*).

To confirm whether the electrical confinement of IPSPs to the dendrites is consistent with our understanding of octopus cell biophysics, we developed an improved biophysically and anatomically accurate computational model of octopus cells based on our findings (*Figure 4—figure supplement 1*; *McGinley et al., 2012*; *Manis and Campagnola, 2018*). Passive and active properties of the model were first aligned with experimental data using a scaling factor (scl), which adjusts the maximum conductance of voltage-gated potassium (Kv) ($\bar{g}_{KLT}$, $\bar{g}_{KHT}$, $\bar{g}_{KA}$) and hyperpolarization-activated cyclic nucleotide-gated (HCN) ($\bar{g}_h$) channels (*Figure 4—figure supplement 1A, B*). To further align the model with experimental data, we adjusted input resistance ($R_N$) and specific membrane resistance ($R_m$) to match experimentally measured input resistance under physiological conditions and with Kv and HCN block (*Figure 4—figure supplement 1D–E*). We further examined the influence of leak, Kv, and HCN conductances on the neuron's input resistance. In a passive scenario, where leak, Kv, and HCN conductances are set to zero ($g_{leak}=0$; $g_{Kv}=0$; $g_h=0$), the input resistance scaled proportionally with specific membrane resistance (*Figure 4—figure supplement 1F*, grey). However, with active leak

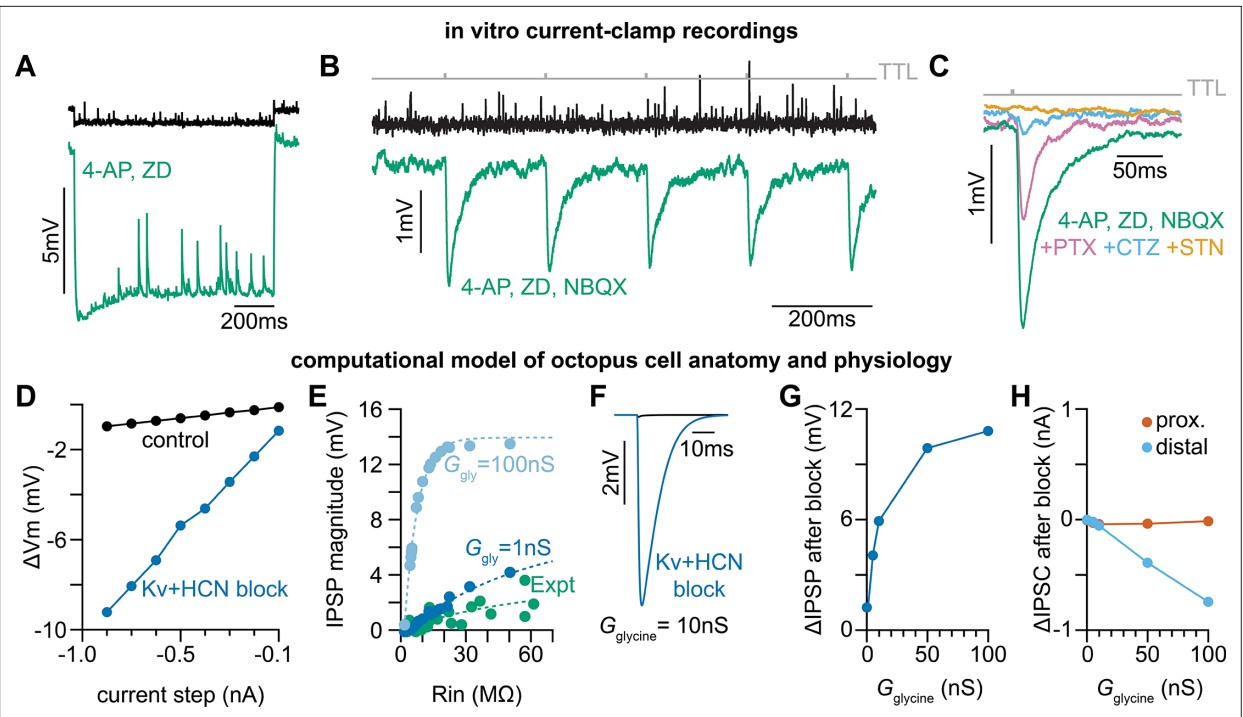

**Figure 4.** Octopus cells receive glycinergic inhibitory post synaptic potentials. (**A**) Voltage responses to a –200 pA current injection. This representative neuron hyperpolarized 0.7 mV (black) in control conditions. After bath application of 100 µM 4-Aminopyridine (4-AP) and 50 µM ZD 7288 (ZD), hyperpolarizing responses to the same –200 pA current injection increased to 8.8 mV at steady state (green). (**B**) Postsynaptic responses to ChR2 stimulation of glycinergic terminals (*Slc6a5*-ChR2) with a 5 Hz train (gray) of 1ms full-field blue light pulses before (black) and after bath application of 100 µM 4-AP, 50 µM ZD, and 15 µM NBQX (green: n=9 cells, 8 mice). Increased input resistance reveals inhibitory potentials that are otherwise difficult to detect. (**C**) Postsynaptic responses to *Slc6a5*-ChR2 stimulation after bath application of 100 µM 4-AP, 50 µM ZD, and 15 µM NBQX (green), with further sequential addition of 20 µM picrotoxin (PTX, pink), 100 µM cyclothiazide (CTZ, blue), and 500 nM strychnine (STN, orange: n=6 cells, 5 mice). (**D**) Change in membrane voltage in response to hyperpolarizing somatic current steps in a morphologically and biophysically realistic model of octopus cells before (black) and after removal of voltage-gated potassium (Kv) and hyperpolarization-activated cyclic nucleotide-gated (HCN) channels (blue). As in in vitro current-clamp recordings (**A**), removing Kv and HCN channels increased the magnitude of voltage responses (ΔVm) to hyperpolarizing current. (**E**) IPSP magnitude in experimental data (green) and the model (dark and light blue) as a function of input resistance. In somatic measurements, IPSP size increases with input resistance. Modeled IPSPs are shown for two conductance levels (1nS, dark blue; 100nS light blue). (**F**) IPSPs measured at the soma of a modeled octopus cell before (black) and after removal of Kv and HCN channels (blue). As in in vitro current-clamp recordings, this allows for somatic IPSP detection. (**G**) Change in the magnitude of soma-measured IPSPs after removal of Kv and HCN channels. IPSP magnitude increases with Kv and HCN block for all glycine conductance levels. (**H**) Change in the magnitude of locally measured IPSCs at proximal (orange) and distal (blue) dendritic locations after removal of Kv and HCN channels. IPSC magnitude at the synapse decreases with Kv and HCN block for all glycine conductance levels. Change in IPSC magnitude is largest on distal dendrites.

The online version of this article includes the following figure supplement(s) for figure 4:

**Figure supplement 1.** Optimizing active and passive properties of an octopus cell model.

conductances, $R_N$ was constrained and showed only a slight increase relative to $R_m$, suggesting attenuation of PSPs during their propagation to the soma (*Figure 4—figure supplement 1G*). As in our current-clamp recordings (*Figure 4A*), removal of Kv and HCN conductances in the model changed the input resistance and current-voltage relationship of the neuron, resulting in reduced electrotonic isolation (*Figure 4—figure supplement 1C*, *Figure 4D*). We next tuned the glycine conductance ($G_{gly}$) to match experimental results (*Figure 4E*). Activation of large (10nS) dendritic glycinergic conductances induced negligible hyperpolarizing voltage changes in the model (*Figure 4F*, black). With increased input resistance and reduced electrotonic isolation after Kv and HCN block, dendritic IPSPs measured at the soma were detectable (*Figure 4F*, blue), consistent with our in vitro recordings.

While blocking Kv and HCN allowed us to reveal IPSPs at the soma, it is possible that reduced electrotonic isolation does not entirely explain the increase in somatically-measured IPSP amplitude. Changes in driving force could increase the magnitude of synaptic currents and therefore increase the magnitude of experimentally measured synaptic potentials. We used the model to explore if

changes in driving force under Kv and HCN block contribute to larger IPSPs recorded at the soma. We simulated dendritic glycinergic conductances over a range of values and in the presence of blocked Kv and HCN channels (*Figure 4—figure supplement 1H–K*). Kv and HCN block and the resulting change in input resistance increased the magnitude of soma-measured IPSPs for all glycine conductances (*Figure 4G*). In contrast, following the pharmacological blockade of Kv and HCN channels, we observed either a reduction in the amplitude of inhibitory postsynaptic currents (IPSCs) at distal synapses (*Figure 4H*, blue) or little change in IPSC amplitude at proximal synapses (*Figure 4H*, orange). These findings indicate that alterations in IPSCs are unlikely to account for the observed increase in inhibitory postsynaptic potential (IPSP) magnitude recorded at the soma after Kv and HCN channel block (*Figure 4—figure supplement 1I–J*). Transfer impedance (ZT), a measure of signal transmission efficiency, was dramatically increased after Kv and HCN block (*Figure 4—figure supplement 1K*). Results from the model provide evidence of functional glycinergic synaptic transmission that is difficult to detect with in vitro somatic recordings due to substantial attenuation of IPSPs as they travel from the dendrites to the soma.

## Inhibition decreases the magnitude and advances the timing of dendritic SGN inputs

SGN synapses onto octopus cell dendrites are arranged tonotopically, with higher frequency SGNs from the base of the cochlea terminating on distal dendrites and lower frequency SGNs from more apical positions terminating proximally (*Figure 1A*). This organization has been proposed to re-synchronize coincidentally firing SGNs that are activated at slightly different times due to the time it takes for the sound stimulus to travel from the base to the apex of the cochlea (*McGinley et al., 2012*). To test how inhibition in the dendrites shapes coincidence detection, we first used our model to explore the influence of simultaneous activation of inhibitory and excitatory synapses at varying locations along the dendritic tree (*Hao et al., 2009*; *Koch et al., 1983*). We modelled how somatically recorded EPSPs are affected by the location of inhibition by moving the site of excitation relative to inhibitory synapses placed either on proximal or distal dendrites (*Figure 5A*, *Figure 5—figure supplement 1*). In our model, inhibitory synapses that are located proximally to excitation (*Figure 5—figure supplement 1A*) had less of an influence on excitation recorded at the soma compared to those inhibitory synapses located distally to excitation (*Figure 5—figure supplement 1B*). We analyzed the effect of inhibitory synapse location on somatically measured EPSPs using varying excitatory and inhibitory synaptic weight values (*Figure 5—figure supplement 1C–F*) and determined that inhibitory conductances ($G_{Gly}$) between 6 and 10 nS produced values within experimental ranges. For both inhibition proximal to excitation (*Figure 5B–C*) and inhibition distal to excitation (*Figure 5D–E*), the model predicted that inhibition reduces EPSP amplitude and accelerates EPSP peak timing at the soma. Thus, the presence of inhibition could modulate EPSP timing in dendritic compartments during continuous auditory stimuli, when inhibition can be recruited after the onset of a sound and thus allow for adaptable temporal processing during the sound's duration.

To directly test if the prediction that temporally coincident excitation and inhibition affects the timing and amplitude of EPSPs as they travel towards the octopus cell soma, we coincidently activated SGNs and glycinergic inputs in vitro. In these experiments, the octopus cell properties were not altered pharmacologically and inhibition was undetectable or only visible with averaging over many sweeps (*Figure 5F*, blue). When synaptic inhibition was evoked together with excitation (*Figure 5F*, green), EPSPs recorded in the soma were smaller and faster than when excitation was evoked alone (*Figure 5F*, black: n=8 cells, 6 mice). ChR2-evoked inhibition decreased EPSP heights by 25.2 ± 9.0% (*Figure 5G*, green) and shifted the peak of EPSPs forward 57.5 ± 26 µs (*Figure 5H*, green). This effect was mimicked by bath application of 25 µM glycine (*Figure 5G–H*, blue: n=4 cells, 3 mice). Further, bath application of 1 µM STN had the opposite effect, resulting in larger EPSPs and delayed peak times (*Figure 5F–H*, orange: n=5 cells, 4 mice). These findings suggest that the timing of EPSP arrival in the soma may be shaped both by tonically active glycine channels and the release of synaptic glycine onto the dendrite. Of note, many SGNs also terminate on the octopus cell soma, where inhibition is minimal. This suggests that the octopus cell's ability to act as a coincidence detector depends on two stages of compartmentalized computations, one in the dendrite that combines excitation and inhibition to provide important information about which frequencies co-occur in a complex sound stimulus and one in the soma that is restricted by the rigid temporal summation window for

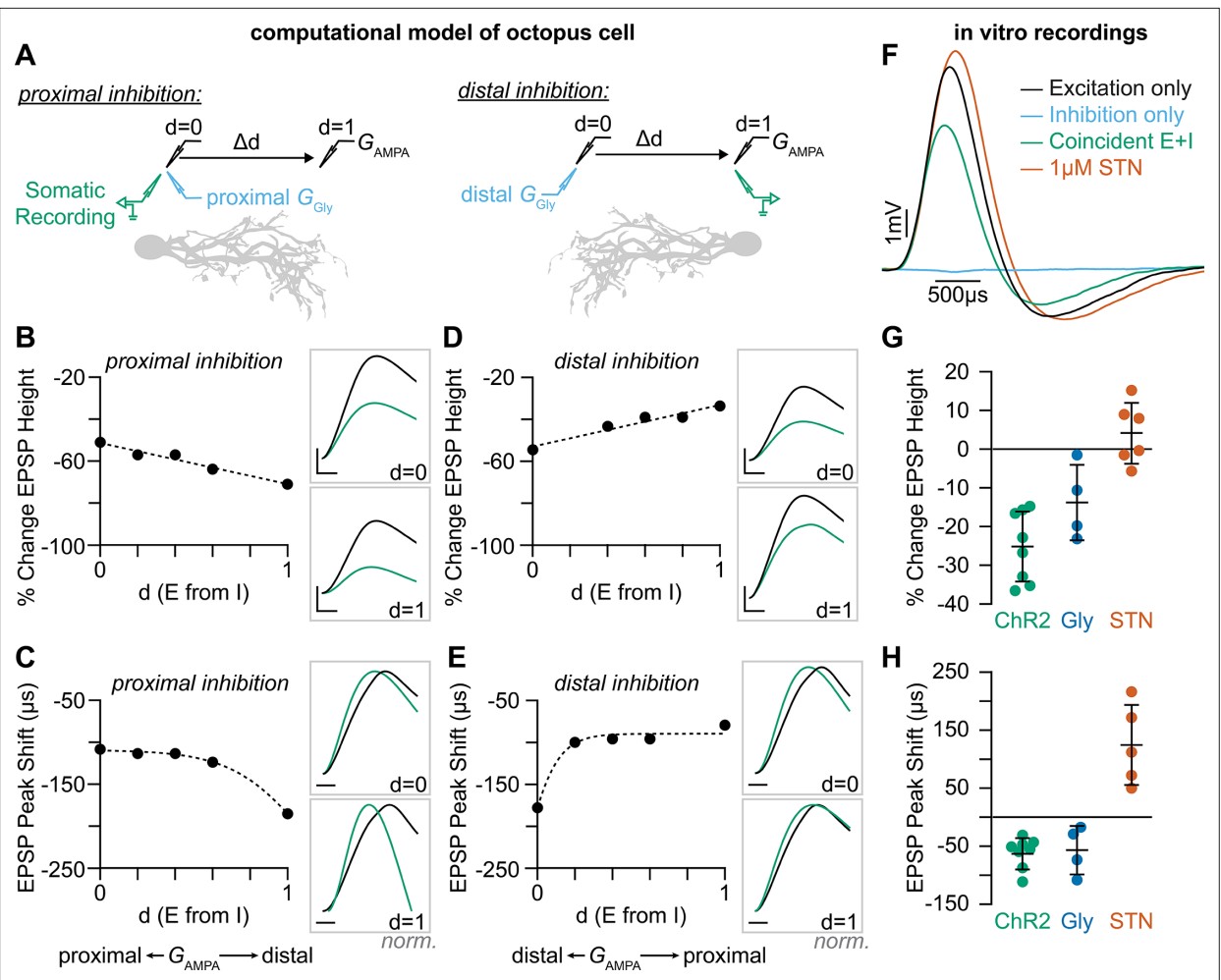

**Figure 5.** Coincident excitation and inhibition on octopus cell dendrites advances EPSP peak times. (**A**) The impact of distance between excitatory and inhibitory synapses was measured in a computational model of octopus cells. Inhibitory synapses were placed either proximally or distally to excitation. Excitatory synapses were placed at varying locations along the dendritic arbor to change the anatomical distance (Δd) where d=0 is the location of inhibition ($G_{Gly}$) and d=1 is the condition where excitation ($G_{AMPA}$) and inhibition are maximally separated. EPSPs were measured at the soma in all conditions (green). (**B–E**) Quantification of the percent change in soma-measured EPSP magnitude (*B, D*; $G_{AMPA}$=5 nS) and the shift in EPSP peak timing (*C, E*; $G_{AMPA}$=2 nS) in models of proximal (**B–C**) and distal (**D–E**) inhibition. Inhibitory conductances ($G_{Gly}$) between 6 and 10 nS produced values within experimental ranges. Example traces show EPSPs with (green) and without (black) inhibition at d=0 and d=1. Distal dendrites (d=1) have higher local input resistance and lower IPSP attenuation due to the sealed end. Inset scale bars are 1 mV, 200 ms. (**F–H**) Coincident stimulation of excitation and inhibition changes EPSP shape. (**F**) Representative responses to independent stimulation of excitatory spiral ganglion neurons (SGNs; black), independent stimulation of inhibitory inputs (light blue), coincident stimulation of both excitation and inhibition (green), and independent stimulation of excitatory SGNs with the addition of 1 µM strychnine (STN). Quantification of the percent change in EPSP height (**G**) and the shift in EPSP peak timing (**H**) during coincident *Slc6a5*-ChR2 activation of inhibitory inputs (green: n=8 cells, 6 mice), bath application of 25 µM glycine (dark blue: n=4 cells, 3 mice), and bath application of 1 µM STN (orange: n=5 cells, 4 mice). Activation of glycinergic receptors during excitation decreased EPSP heights and advanced EPSP peaks. Blocking of tonically active glycine receptors slowed and delayed EPSPs. Data are presented as mean ± SD. Markers represent the average quantification for a cell.

The online version of this article includes the following source data and figure supplement(s) for figure 5:

**Source data 1.** Data included in *Figure 5G*.

**Source data 2.** Data included in *Figure 5H*.

**Figure supplement 1.** Impact of inhibitory synaptic location and distance between excitatory and inhibitory synapse on somatic EPSP amplitude and timing.

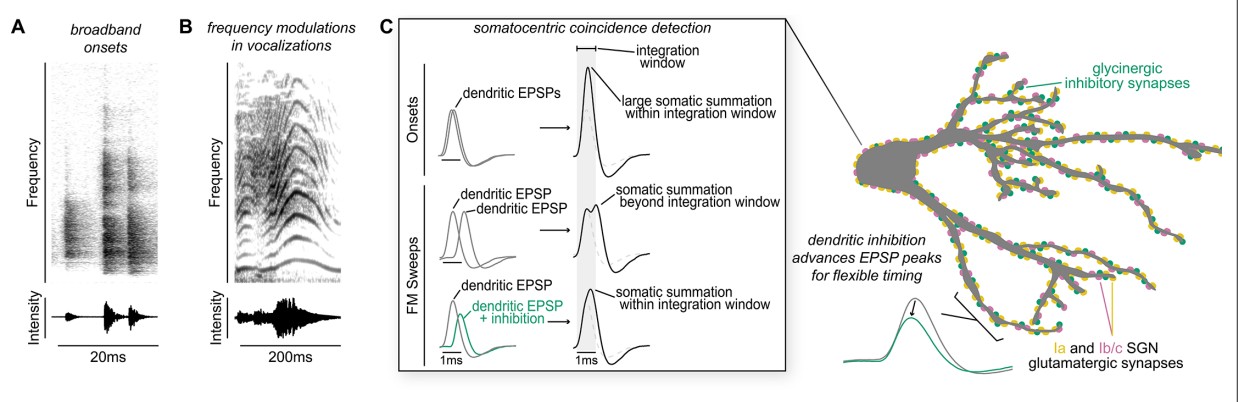

**Figure 6.** Proposed model of flexible dendritic timing for precise somatic coincidence detection. (**A–B**) Spectrograms show how co-occurring frequencies in broadband onsets, shown for a clicking sound (**A**), and frequency modulations, shown for human squeaking (**B**), change in their strength (top) and total intensity (bottom) on different time scales. Despite the timing differences, both kinds of stimuli are hypothesized to result in somatic summation during the octopus cell's spike integration window. (**C**) Excitation must summate at the soma of octopus cells within a narrow time window (~1ms) to achieve a depolarization rate rapid enough to trigger action potentials. Summation of excitation alone accounts for responses to sound onsets (inset, top). During stimuli that require summation over a longer time period, such as frequency modulated sweeps, synaptic inhibition can accelerate EPSPs as they travel along octopus cell dendrites towards the soma for coincidence detection computations (inset, bottom). We propose a mechanism for preferential processing of a subset of excitatory inputs where selective temporal advancement of a subset of EPSPs by local inhibition could expand the effective window for coincidence detection at the soma.

coincidence detection. Together with the electrotonic properties of the octopus cell and the dominance of low threshold, low jitter Ia SGN inputs, these combined computations can enable reliable coincidence detection and cross-frequency binding needed for perception of sound.

## Discussion

Coincidence detection plays a critical role in many cognitive and perceptual processes, from the ability to localize sound to the binding of auditory and visual features of a common stimulus. Depending on the computation, the temporal window for integration can range widely, thereby requiring circuitry with distinct anatomical and physiological properties. Here, we describe a two-domain mechanism for detecting co-occurring frequencies with different degrees of precision. By mapping and selectively activating synaptic inputs onto octopus cells both in vitro and in a computational model, we revealed that compartmentalized dendritic nonlinearities impact the temporal integration window under which somatic coincidence detection computations are made. The arrival of many small, reliable excitatory inputs (*Figure 3*) from low-threshold SGNs (*Figure 2*) is continuous throughout an ongoing stimulus. We demonstrate that glycinergic inhibition to octopus cell dendrites (*Figure 1*) can shift the magnitude and timing of SGN EPSPs as they summate in the soma (*Figure 4*, *Figure 5*). The narrow window for coincidence detection computations allows the octopus cell to respond with temporal precision using momentary evidence provided by SGNs at the onset of the stimulus. We propose that, as a stimulus persists, inhibition onto octopus cell dendrites can adjust EPSPs before they arrive at the soma for the final input-output computation. This allows the cell to make an additional computation with a slightly longer window for evidence accumulation without compromising the accuracy of the computation in the soma (*Figure 6*).

As coincidence detectors in the auditory system, octopus cells are faced with the challenge of recognizing complex sounds that include many frequencies that co-occur from the beginning to the end of the stimulus. As shown by in vivo recordings (***Recio-Spinoso and Rhode, 2020***; ***Lu et al., 2022***), octopus cells respond well to cues that include complex spectrotemporal patterns, including frequency modulations beyond the onset of the stimulus (***Lu et al., 2022***). Given that the auditory environment is filled with overlapping sound stimuli, such responses presumably allow the octopus cell to encode which frequencies modulate together. Our data thus support the role of octopus cells beyond simple onset coincidence detectors that rely solely on the temporal summation of excitation. The results suggest that, in addition to high Kv and HCN conductances at rest, the addition of dendritic

inhibition transforms the magnitude and timing of SGN signals as they arrive in the cell body, which may expand the response selectivity of an octopus cell and allow them to become a slightly leakier integrator that can accumulate evidence beyond onsets. Although, this inhibition is difficult to detect because of shunting, our data demonstrate that it is both present and impactful.

As well as needing to work beyond onsets, an effective coincidence detector in the auditory system must also function reliably across a range of sound intensities. Intensity information is encoded by the number and types of SGNs that are activated in the cochlea. The Ia, Ib, and Ic molecular subtypes defined in mouse (*Petitpré et al., 2018*; *Shrestha et al., 2018*; *Sun et al., 2018*) broadly correspond to the anatomically and physiologically defined subtypes described across species (*Siebald et al., 2023*; *Moser et al., 2023*). We find that the majority of inputs onto octopus cells come from Ia SGNs, which most closely correspond to the low-threshold, high-spontaneous rate population. Consistent with this result, single-unit SGN recordings in cats demonstrated a bias towards low-threshold, high-spontaneous rate axon collaterals in the octopus cell area (*Liberman, 1993*). Low-threshold SGNs are also characterized by short first spike latencies and low temporal jitter (*Buran et al., 2010*; *Oliver et al., 2006*; *Bourien et al., 2014*; *Costalupes et al., 1984*). A hallmark of the octopus cell is the fact that it only fires action potentials when many SGN inputs are activated within a narrow period of time (*McGinley and Oertel, 2006*). The presence of many low-threshold and temporally precise inputs on the octopus cell may help ensure that coincidence detection still works reliably for quiet sounds. Further, Ia inputs onto octopus cells do not exhibit paired-pulse depression, similar to low levels of depression seen in Ia inputs to bushy cells (*Zhang et al., 2022*). The presence of SGN inputs without paired-pulse depression could be beneficial for encoding sustained auditory signals. Finally, while Ia SGNs are over-represented, Ib and Ic inputs are also present. Since precise, low-threshold SGN responses can be saturated by background noise such that responses to relevant stimuli are masked (*Liberman, 1978*; *Costalupes et al., 1984*; *Bourien et al., 2014*; *Huet et al., 2016*), recruitment of higher threshold SGNs at higher sound intensities may compensate for this tradeoff.

The presence of inhibitory inputs onto dendrites is a fundamental feature of the nervous system and, in other systems, contributes to a neuron's final computation. For example, direction selectivity computations in dendrites of retinal cells require excitation-inhibition interactions in dendritic compartments (*Ding et al., 2016*; *Stasheff and Masland, 2002*). In pyramidal cells of the cortex and hippocampus, the spatial distribution of inhibition impacts dendritic non-linearities in a branch-selective manner (*Bloss et al., 2016*; *Gidon and Segev, 2012*; *Iascone et al., 2020*; *Jadi et al., 2012*; *Lovett-Barron et al., 2012*; *Hao et al., 2009*). However, octopus cells do not share all mechanisms for dendritic computation seen in cortical and hippocampal neurons. Resting conductances and low-threshold potassium conductances may suppress voltage-gated calcium spikes and attenuate the magnitude of action potentials as they backpropagate through the soma and dendritic tree (*Golding et al., 1999*; *Cao and Oertel, 2005*; *Bal and Baydas, 2009*; *Bal and Oertel, 2007*). Therefore, subthreshold dendritic integration of coincident excitation and local inhibition may be the primary computation that occurs in octopus cell dendrites before action potential generation near the soma.

Although this work uncovers a role for inhibition, a deeper understanding of octopus cell computations will require determining what information is carried by inhibitory inputs. Despite the well-established presence of presynaptic glycinergic puncta in the octopus cell area (*Kemmer and Vater, 1997*; *Kolston et al., 1992*; *Juiz et al., 1996*; *Moore et al., 1996*) and glycinergic receptor expression in octopus cells (*Friauf et al., 1997*; *Sato et al., 2000*; *Schofield and Cant, 1996*; *Thompson et al., 1985*; *Adams and Mugnaini, 1987*), it is unknown where glycinergic innervation originates. Local neurons within the CNC could provide inhibition; however it remains unclear whether octopus cells receive connections from D-Stellate (*Oertel et al., 1990*), L-Stellate (*Ngodup et al., 2020*), or tuberculoventral cells (*Wickesberg et al., 1991*). Outside of the CNC, terminal degeneration experiments in cats suggested the superior periolivary nucleus (SPON) and the ventral division of the lateral lemniscus (VNLL) as potential sources of descending inhibition to the octopus cell area (*Kane, 1977*). Octopus cells provide excitatory input to both the SPON (*Friauf and Ostwald, 1988*; *Schofield, 1995*; *Thompson and Thompson, 1991*; *Zook and Casseday, 1985*; *Felix Ii et al., 2017*) and the VNLL (*Schofield and Cant, 1997*; *Smith et al., 2005*; *Berger et al., 2014*; *Nayagam et al., 2005*; *Adams, 1997*; *Vater and Feng, 1990*), raising the possibility of feedback inhibition from the auditory brainstem as a circuit mechanism that elongates temporal summation windows during ongoing stimuli. Such descending feedback inhibition is not rapid enough to prevent or alter the characteristic

octopus cell onset response, but could change the effective coincidence detection window as the stimulus continues or limit the duration of the response. Future studies will be required to identify the source of inhibition and its organization within dendritic compartments. If inhibitory inputs tonotopically match the local, narrowly-tuned dendritic SGN inputs, it is possible that frequency-matched inhibition could influence spectral selectivity or feature extraction. On the other hand, broadly tuned inhibition could reduce depolarization block or serve as a temporal milestone that signals gaps or offsets. Further characterization of in vivo octopus cell responses in complex sound environments may clarify the effect of noise on signal detection and reveal additional features of this cell's contributions to perception of the auditory world.

# Materials and methods

**Key resources table**

| Reagent type (species) or resource | Designation | Source or reference | Identifiers | Additional information |
|---|---|---|---|---|
| Biological sample (*Mus musculus*) | *Foxg1*^Cre/+: *Foxg1*^tm1.1(cre)Ddmo | Jackson Laboratory | RRID:IMSR_JAX:029690 | |
| Biological sample (*Mus musculus*) | *Foxg1*^tm1.1Fsh | *Miyoshi and Fishell, 2012* | MGI:5441367 | |
| Biological sample (*Mus musculus*) | Tg(EIIa-cre)C5379Lmgd | Jackson Laboratory | RRID:IMSR_JAX:003314 | |
| Biological sample (*Mus musculus*) | *Ntng1*^Cre/+: *Ntng1*^em1(cre)Kfra | *Bolding et al., 2020* | MGI:6740547 | |
| Biological sample (*Mus musculus*) | *Myo15*^iCre/+: *Myo15a*^tm1.1(cre)Ugds | *Caberlotto et al., 2011* | MGI:4361284 | |
| Biological sample (*Mus musculus*) | *Slc6a5*^Cre/+: *Slc6a5*^tm1.1(cre)Ksak | *Kakizaki et al., 2017* | MGI:6382286 | |
| Biological sample (*Mus musculus*) | Ai14: *Gt(ROSA)26Sor*^tm14(CAG-tdTomato) | Jackson Laboratory | RRID:IMSR_JAX:007914 | |
| Biological sample (*Mus musculus*) | Ai34: *Gt(Rosa)26Sor*^tm34.1(CAG-Syp/tdTomato) | Jackson Laboratory | RRID:IMSR_JAX:012570 | |
| Biological sample (*Mus musculus*) | RCE:FRT: *Gt(Rosa)26Sor*^tm1.2(CAG-EGFP)Fsh | *Sousa et al., 2009* | RRID:MMRRC_032038-JAX | |
| Biological sample (*Mus musculus*) | Ai32: *Gt(Rosa)26Sor*^tm32(CAG-COP4*H134R.EYFP) | Jackson Laboratory | RRID:IMSR_JAX:01256 | |
| Biological sample (*Mus musculus*) | *Thy1*: Tg(Thy1-YFP)HJrs | Jackson Laboratory | RRID:IMSR_JAX:003782 | |
| Antibody | Chicken polyclonal anti-GFP | Aves Labs | Cat# GFP-1020; RRID:AB_10000240 | 1:1000 |
| Antibody | Rabbit polyclonal anti-RFP | Rockland | Cat# 600-401-379; RRID:AB_2209751 | 1:1000 |
| Antibody | Goat polyclonal anti-Calretinin | Swant | Cat# CG1; RRID:AB_1000034 | 1:1000 |
| Antibody | Guinea Pig polyclonal anti-VGlut1 | Synaptic Systems | Cat# 135 304; RRID:AB_887878 | 1:500 |
| Chemical compound, drug | 4-Amynopyridine | Tocris | CAS: 504-24-5 | |
| Chemical compound, drug | 2,3-dioxo-6-nitro-7-sulfamoyl-benzo[f]quinoxaline (NBQX) | Tocris | CAS: 479347-86-9 | |
| Chemical compound, drug | ZD 7288 | Tocris | CAS: 133059-99-1 | |
| Chemical compound, drug | Cyclothiazide | Tocris | CAS: 2259-96-3 | |

*Continued on next page*

*Continued*

| Reagent type (species) or resource | Designation | Source or reference | Identifiers | Additional information |
|---|---|---|---|---|
| Chemical compound, drug | Strychnine | Sigma Aldrich | S0532, CAS: 57-24-9 | |
| Software, algorithm | Imaris | Oxford Instruments | Imaris | |
| Software, algorithm | ImageJ | https://imagej.net/ij/ | ImageJ | |
| Software, algorithm | pClamp9 | Molecular Devices | pClamp9 | |
| Software, algorithm | Igor Pro | Wavemetrics | Igor Pro | |
| Software, algorithm | NeuroMatic | http://www.neuromatic.thinkrandom.com/ | NeuroMatic | |
| Software, algorithm | NEURON simulation environment | https://neuron.yale.edu/neuron/ | NEURON simulation environment | |
| Software, algorithm | NEURON simulation code | https://modeldb.science/2018259 | NEURON simulation code | |
| Software, algorithm | Imaris | Oxford Instruments | Imaris | |

## Animal use and transgenic mouse lines

All procedures were approved by and conducted in accordance with Harvard Medical School Institutional Animal Care and Use Committee (Protocol Number IS00000067-9). Male and female mice (*Mus musculus*) were bred on a C57BL/6 background at the Harvard Center for Comparative Medicine or obtained from Jackson Laboratories. Mice were housed in groups of up to five animals and maintained on a 12 hr light/dark cycle. Transgenic alleles were heterozygous for each transgene for all experimental animals. Descriptions of allele combinations for all experiments can be found in *Supplementary file 1*.

Spiral ganglion neurons (SGNs) and their central axons, that is auditory nerve fibers, were targeted using either *Foxg1*$^{tm1.1(Cre)Ddmo}$ (*Foxg1*$^{Cre}$; *Kawaguchi et al., 2016*) or *Foxg1*$^{Flp}$, both of which drive robust reporter expression in neurons in the auditory and vestibular ganglion (*Pauley et al., 2006*; *Hatini et al., 1999*) and the neocortex (*Hanashima et al., 2002*; *Tao and Lai, 1992*), but not in brainstem or midbrain neurons. *Foxg1*$^{Flp}$ mice were generated by crossing the *Foxg1*$^{tm1.1Fsh}$ mouse line (*Miyoshi and Fishell, 2012*) with the Tg(EIIa-Cre)C5379Lmgd mouse line (*Lakso et al., 1996*), then backcrossing to isolate the Flp transgene and remove the Cre transgene.

Inhibitory inputs to octopus cells were targeted with *Slc6a5*$^{tm1.1(Cre)Ksak}$ mice (*Slc6a5*$^{Cre}$; *Kakizaki et al., 2017*).

Octopus cells were sparsely labeled with the Tg(Thy1-YFP)HJrs (*Thy1*) mouse line (*Feng et al., 2000*). This line labels ~0–15 octopus cells amongst other neurons throughout the brain.

Ib/c SGNs were targeted using the *Ntng1*$^{em1(Cre)Kfra}$ (*Ntng1*$^{Cre}$) mouse line, which drives expression in neurons throughout the nervous system (*Figure 2—figure supplement 1D–E*) and disrupts expression of the endogenous allele (*Bolding et al., 2020*). Auditory brainstem responses in adult *Ntng1*$^{Cre/+}$ mice are normal. Ic SGNs were sparsely targeted with the *Myo15a*$^{tm1.1(Cre)Ugds}$ (*Myo15*$^{iCre}$) mouse line (*Caberlotto et al., 2011*).

Fluorescent reporters included *Gt(Rosa)26Sor*$^{tm14(CAG-tdTomato)}$ (Ai14, tdT; *Madisen et al., 2010*), *Gt(Rosa)26Sor*$^{tm34.1(CAG-Syp/tdTomato)}$ (Ai34, syp/tdT), and *Gt(Rosa)26Sor*$^{tm1.2(CAG-EGFP)Fsh}$ (RCE:FRT, EYFP; *Sousa et al., 2009*). We also used *Gt(Rosa)26Sor*$^{tm32(CAG-COP4*H134R.EYFP)}$ (Ai32, ChR2; *Madisen et al., 2012*) to drive synaptic activity in in vitro slice experiments.

## Histology and reconstructions

For immunohistochemical labeling, mice were deeply anesthetized with isoflurane and transcardially perfused with 15 mL of 4% paraformaldehyde (PFA) in 0.1 M phosphate-buffered saline (PBS) using a peristaltic pump (Gilson). Whole skulls containing brain and cochlea were immediately transferred to 20 mL of 4% PFA and post-fixed overnight at 4 °C. Fixed brains and cochlea were removed from the skulls and washed with 0.1 M PBS.

## Brain sections

Brains were collected from mice of both sexes, aged 28–38 days, and embedded in gelatin-albumin hardened with 5% glutaraldehyde and 37% PFA (*Connelly et al., 2017*). Sections were cut at 35, 65,

or 100 μm with a vibrating microtome (Leica VT1000S) and free-floating tissue was collected in 0.1 M PBS. For sections less than 65 μm, tissue was permeabilized and nonspecific staining was blocked in a solution of 0.2% Triton X-100 and 5% normal donkey serum (NDS, RRID:AB_2337258) in 0.1 M PBS for 1 hr. After blocking, tissue was treated with primary antibody in a solution containing 0.2% Triton X-100 and 5% NDS in PBS for 1–2 nights at room temperature. Primary antibodies used were: chicken anti-GFP (1:1000, RRID:AB_10000240), rabbit anti-RFP (1:1000, RRID:AB_2209751), goat anti-calretinin (1:1000, RRID:AB_1000034), and guinea pig anti-VGLUT1 (1:500, RRID:AB_887878). Sections were washed in 0.1 M PBS then incubated in a secondary antibody solution (1:1000) containing 0.2% Triton X-100 and 5% NDS for 2–3 hr at room temperature. Tissue sections were mounted on charged slides and coverslipped (Vectashield Hardset Antifade Mounting Medium with DAPI), and imaged using a Zeiss Observer.Z1 confocal microscope.

For 100 μm sections, tissue was washed in CUBIC-1A solution (*Susaki et al., 2015*) for 1 hr for strong permeabilization and delipidization (*Matsumoto et al., 2019*; *Susaki et al., 2015*). Tissue was then further permeabilized and nonspecific staining was blocked in a solution of 0.2% Triton X-100 and 5% NDS in 0.1 M PBS for 1 hr. After blocking, tissue was treated with primary antibody in a solution containing 0.2% Triton X-100 and 5% NDS in PBS for 4 nights at 37 °C. Primary antibodies used were: chicken anti-GFP (1:1000, RRID:AB_10000240), and rabbit anti-RFP (1:1000, RRID:AB_2209751). Sections were then incubated in a secondary antibody solution (1:400) containing 0.2% Triton X-100 and 5% NDS for 4 nights at 37 °C. Tissue sections were pre-incubated in CUBIC2 solution, then temporarily mounted on uncharged slides with CUBIC2 solution for immediate imaging using a Zeiss Observer.Z1 confocal microscope.

### 3D reconstructions

Octopus cells and synaptic puncta were reconstructed in Imaris (Oxford Instruments). YFP signal from the target octopus cell was used to generate a surface reconstruction and mask syp/tdT signal. Dendrites were reconstructed using the masked YFP signal and separated into 10 μm increments. Masked syp/tdT puncta were marked and localized to a 10 μm increment of the dendritic tree. Synapse counts, dendrite metrics, and masked channels were exported to Excel (Microsoft) for further analysis.

### Cochlea sections

Cochlea were collected from mice of both sexes, aged 28–42 days. The bony labyrinth of the inner ear was decalcified in 0.5 M ethylenediamine tetraacetic acid (EDTA) for 3 nights at 4 °C and embedded in gelatin-albumin hardened with 5% glutaraldehyde and 37% PFA. Sections were cut at 65 μm with a vibrating microtome (Leica VT1000S) and free-floating tissue was collected in 0.1 M PBS. Sections were washed in CUBIC-1A solution for 1 hr for strong permeabilization and delipidization. Tissue was further permeabilized and nonspecific staining was blocked in a solution of 0.2% Triton X-100 and 5% NDS in 0.1 M PBS for 1 hr. After blocking, tissue was treated with primary antibody in a solution containing 0.2% Triton X-100 and 5% NDS in PBS for 2 nights at room temperature. Primary antibodies used were: chicken anti-GFP (1:1000, RRID:AB_10000240), rabbit anti-RFP (1:1000, RRID:AB_2209751), goat anti-calretinin (1:1000, RRID:AB_1000034). Sections were then incubated in a secondary antibody solution (1:500) containing 0.2% Triton X-100 and 5% normal goat serum for 2–3 hr at room temperature. Tissue sections were mounted on charged slides, coverslipped (Vectashield Hardset Antifade Mounting Medium with DAPI), and imaged using a Zeiss Observer.Z1 confocal microscope.

### Acute slice electrophysiology

Data were obtained from mice of both sexes, aged 24–47 days. Mice were deeply anesthetized with isoflurane and perfused transcardially with 3 mL of 35 °C artificial cerebral spinal fluid (ACSF; 125 mM NaCl, 25 mM glucose, 25 mM $NaHCO_3$, 2.5 mM KCl, 1.25 mM $NaH_2PO_4$, 1.4 mM $CaCl_2$, and 1.6 mM $MgSO_4$, pH adjusted to 7.45 with NaOH). For high calcium concentration experiments presented in *Figure 3C*, ACSF contained 125 mM NaCl, 25 mM glucose, 25 mM $NaHCO_3$, 2.5 mM KCl, 1.25 mM $NaH_2PO_4$, 2.4 mM $CaCl_2$, and 1.3 mM $MgSO_4$. Mice were rapidly decapitated and the brain was removed and immediately submerged in ACSF. Brains were bisected and 250 μm slices were prepared in the sagittal plane with a vibrating microtome (Leica VT1200S; Leica Systems). Prepared slices were incubated for 30 min at 35 °C, then allowed to recover at room temperature for at least 30 min. ACSF was continuously bubbled with 95% $O_2$/5% $CO_2$.

Whole-cell recordings were conducted at 35 °C using a Multiclamp 700B (Molecular Devices) in current-clamp mode with experimenter adjusted and maintained bridge balance and capacitance compensation. Data were filtered at 12 kHz, digitized at 83–100 kHz, and acquired using pClamp9 (Molecular Devices). Neurons were visualized using infrared Dodt gradient contrast (Zeiss Examiner. D1; Zeiss Axiocam 305 mono). Glass recording electrodes (3–7 MΩ) were wrapped in parafilm to reduce capacitance and filled with an intracellular solution containing 115 mM K-gluconate, 4.42 mM KCl, 0.5 mM EGTA, 10 mM HEPES, 10 mM $Na_2$Phosphocreatine, 4 mM MgATP, 0.3 mM NaGTP, and 0.1% biocytin, osmolality adjusted to 300 mmol/kg with sucrose, pH adjusted to 7.30 with KOH. All membrane potentials are corrected for a –11 mV junction potential.

For optogenetic activation, full-field 475 nm blue light was presented through a 20x immersion objective (Zeiss Examiner.D1). Onset, duration, and intensity of light was controlled by a Colibri5 LED Light Source (Zeiss). Light intensity at the focal plane ranged between 1.9 and 4.1 $mW/mm^2$, corresponding to 6% and 10% intensity on the Colibri5 system. For electrical stimulation, glass stimulating electrodes were placed in the auditory nerve root and 20μs current pulses were generated with a DS3 current stimulator (Digitimer). Light or electrical stimulation intensity was adjusted during the experiment to evoke subthreshold EPSPs, not spikes. During analysis, EPSPs and spikes were distinguished by screening all events with a phase plot analysis. Only stimulation events that evoked EPSPs unambiguously were included for analysis. When presenting electrical and light stimulation together, a series of stimulation pairings with shuffled onset timings was presented to account for cell to cell variability in EPSP and IPSP timings. Data presented is for the stimulation pairings that evoked a maximal shift in EPSP timings.

## Analysis and statistical tests

Cell counts and habenula measurements were performed in ImageJ/FIJI software (National Institutes of Health). Electrophysiology data were analyzed using custom scripts and NeuroMatic analysis routines (*Rothman and Silver, 2018*) in Igor Pro (Wavemetrics).

For data with equal variance (Levene's test), one-way ANOVAs with Tukey's HSD post hoc test were used where appropriate to determine statistical significance. For data with non-homogenous variances, one-way ANOVAs with a Welch F test were used with a Tukey's HSD post hoc test. Errors and error bars report standard deviation (SD) or standard error of the mean (SEM) as noted in figure legends and throughout the text.

## Computational modelling

Computer simulations were performed using the NEURON 8.2 simulation environment (*Hines and Carnevale, 1997*), with an integration time constant of 25μs. The morphology of the octopus neuron was obtained from *McGinley et al., 2012*. The active and passive properties of the model were optimized to match the experimental recordings. We set the passive parameters as follows: internal or axial resistance ($R_i$ or $R_a$) to 150 Ω.cm, membrane resistance ($R_m$) to 5 $KΩ.cm^2$, capacitance ($C_m$) to 0.9 $μF/Cm^2$ and resting membrane potential ($V_m$) to –65 mV. Ion-channel kinetics, maximum conductance densities, Q10 (3), and temperature (22 °C) were obtained from and matched to *Manis and Campagnola, 2018* and the maximal conductances were adjusted using a scaling factor (scl) to align qualitatively with experimental data (*Figure 4—figure supplement 1*): fast $Na^+$ ($\bar{g}_{Na}$=0.83), fast transient $K^+$ ($\bar{g}_{KA}$=0.07), high threshold $K^+$ ($\bar{g}_{KHT}$=0.1875), low-voltage activated $K^+$ ($\bar{g}_{KLT}$=0.75), hyperpolarization-activated cyclic nucleotide-gated (HCN; $\bar{g}_h$=0.07) and leak $K^+$ ($\bar{g}_{leak}$=0.0017). All conductances were uniformly distributed across dendrites and soma, except for HCN conductance, which was only present in dendrites. A baseline scaling factor of 1 was applied under control conditions and 0 under Kv+HCN block conditions. AMPA conductance ($G_{AMPA}$) was set to 5nS to align with experimental data (*Figure 5F*), and glycine conductance ($G_{Gly}$) was set to 1nS to match experimental observations (*Figure 4E*). Reversal potentials for HCN, $Na^+$, and $K^+$ respectively were (in mV), $E_h$=–38, $E_{Na}$=50 and $E_K$=–70. Excitatory AMPA synaptic conductance and inhibitory glycine synaptic conductance were introduced in the proximal and distal dendrites to test the impact of dendritic inhibition on the EPSP height and peak time. The magnitudes of synaptic conductances were tuned to fall within the range seen during experimental data collection. The rise and decay time of AMPA and glycine conductances were tuned to 0.3ms and 3ms respectively, to match experimental data. The reversal potential of AMPA and glycine conductance was set to 0 mV and –80 mV, respectively.

## Acknowledgements

We thank Nace Golding (University of Texas at Austin), Matthew McGinley (Baylor College of Medicine), Philip Joris (KU Leuven), and Bernardo Sabatini (Harvard Medical School) for helpful discussions and feedback. Sadie Quinn, Lucy Lee, and Ryan Merrow provided valuable technical assistance. Bruce Bean (Harvard Medical School) generously provided access to electrophysiological equipment. $Slc6a5^{tm1.1(Cre)Ksak}$, $Ntng1^{em1(Cre)Kfra}$, and $Myo15a^{tm1.1(Cre)Ugds}$ mice were kindly provided by Wade Regehr (Harvard Medical School), Fan Wang (Massachusetts Institute of Technology), and Stefan Heller (Stanford). We thank Rigoberto Ramirez, Tenzin Paljorwa, and Edgar Ramirez for animal care support. We are grateful to the Neurobiology Imaging Facility (NIF) for software availability and to the HMS Research Instrumentation Core for the design and fabrication of temperature regulation equipment. This work was supported by grants from the BRAIN Initiative 1R01NS118402 to LVG, the National Institute on Deafness and Other Communication Disorders 5R01DC009223 to LVG and 1F32DC020070 to LJK, the William Randolph Hearst Fund to LJK, and the Broad Institute's Stanley Center for Psychiatric Research to SH and GF.

## Additional information

### Funding

| Funder | Grant reference number | Author |
| --- | --- | --- |
| National Institute of Neurological Disorders and Stroke | R01NS118402 | Lisa Goodrich |
| National Institute on Deafness and Other Communication Disorders | R01DC009223 | Lisa Goodrich |
| National Institute on Deafness and Other Communication Disorders | F32DC020070 | Lauren J Kreeger |
| William Randolph Hearst Foundation | | Lauren J Kreeger |
| Stanley Center for Psychiatric Research, Broad Institute | | Suraj Honnuraiah Gordon Fishell |

The funders had no role in study design, data collection and interpretation, or the decision to submit the work for publication.

### Author contributions

Lauren J Kreeger, Conceptualization, Formal analysis, Funding acquisition, Investigation, Visualization, Writing – original draft, Writing – review and editing; Suraj Honnuraiah, Formal analysis, Visualization, Methodology, Writing – review and editing; Sydney Maeker, Siobhan Shea, Investigation; Gordon Fishell, Resources, Supervision, Funding acquisition, Writing – review and editing; Lisa Goodrich, Conceptualization, Supervision, Funding acquisition, Writing – original draft, Project administration, Writing – review and editing

### Author ORCIDs

Lauren J Kreeger ⓘ https://orcid.org/0000-0001-9112-1489
Gordon Fishell ⓘ https://orcid.org/0000-0002-9640-9278
Lisa Goodrich ⓘ https://orcid.org/0000-0002-3331-8600

### Ethics

All procedures were approved by and conducted in accordance with Harvard Medical School's Institutional Animal Care and Use Committee (Protocol Number IS00000067-9). For tissue collection, animals were deeply anesthetized with isoflurane and every effort was made to minimize suffering.

Reviewer #1 (Public review): https://doi.org/10.7554/eLife.100492.3.sa1
Reviewer #2 (Public review): https://doi.org/10.7554/eLife.100492.3.sa2
Author response https://doi.org/10.7554/eLife.100492.3.sa3

## Additional files

### Supplementary files

Supplementary file 1. Summary and description of experimental genotypes presented in figures. Experimental genotypes, their abbreviations, and a description of reporter expression are listed for the relevant figures.

MDAR checklist

### Data availability

Data generated during this project are provided in the manuscript and supporting files. Reconstructions are available upon request. Modelling code was uploaded to ModelDB and available at: https://modeldb.science/2018259.

The following dataset was generated:

| Author(s) | Year | Dataset title | Dataset URL | Database and Identifier |
|---|---|---|---|---|
| Honnuraiah S | 2025 | Octopus neuron (Kreeger et al., 2025) | https://modeldb.science/2018259 | ModelDB, 2018259 |

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
