## [Editor Report · eLife Assessment]

Kreeger et al. **convincingly** demonstrate that octopus cells in the mouse cochlear nucleus, previously thought to rely primarily on excitatory inputs for coincidence detection, also receive glycinergic inhibitory synaptic inputs that influence their synaptic integration. Using advanced techniques, including genetic mouse models, optogenetics, microscopy, slice physiology, and computational modeling, this **important** study reveals that inhibition can shunt synaptic currents and alter the timing of dendritic EPSPs, both of which are significant for auditory processing. This research broadens the understanding of octopus cells' roles in sensory processing, highlighting the importance of inhibitory inputs in shaping fast, high-frequency neural response capabilities.

---

## [Referee Report · Reviewer #1 (Public review)]

Kreeger and colleagues have explored the balance of excitation and inhibition in the cochlear nucleus octopus cells of mice using morphological, electrophysiological and computational methods. On the surface, the conclusion, that synaptic inhibition is present, does not seem like a leap. However, the octopus cells have been in the past portrayed as lacking synaptic inhibition. This view was supported by the paucity of glycinergic fibers in the octopus cell area and the lack of apparent IPSPs. Here, Kreeger et al., used beautiful immunohistochemical and mouse genetic methods to quantify the inhibitory and excitatory boutons over the complete surface of individual octopus cells and further analyzed the proportions of the different subtypes of spiral ganglion cell inputs. I think the analysis of synaptic distribution and the origin of the excitatory inputs stands as one of the most complete descriptions of any neuron, leaving little doubt about the presence of glycinergic boutons.

Kreeger et al then examined inhibition physiologically. Recordings from these neurons are notoriously difficult to make because of the enormous leak currents that shunt membrane stimuli and currents, and complicate voltage clamp. The authors have tried to overcome these limitations using drugs to block leak conductances, and computational approaches based on realistic parameters. They conclude that dendritic inhibition can modify the size and kinetics of excitatory signals, and may play out in computations made on temporally dispersed stimuli as might be experienced during a ramp in sound frequency or complex natural sounds like vocalizations.

---

## [Referee Report · Reviewer #2 (Public review)]

Kreeger et.al provided mechanistic evidence for flexible coincidence detection of auditory nerve synaptic inputs by octopus cells in the mouse cochlear nucleus. The octopus cells are highly specialized neurons that, with appropriate stimuli, can fire repetitively at very high rates (> 800 Hz in vivo), yield responses dominated by the onset of sound for simple stimuli, and integrate auditory nerve inputs over a wide frequency span. Previously, it was thought that octopus cells received little inhibitory input, and their integration of auditory input depended principally on temporally precise coincidence detection of excitatory auditory nerve inputs, coupled with a low input resistance established by high levels of expression of certain potassium channels and hyperpolarization-activated channels.

This study provides convincing evidence that octopus cells do in fact receive glycinergic synaptic input that can influence the efficacy of excitatory dendritic synaptic activity. By coupling selected genetic mouse models to characterize synaptic inputs and enable optogenetic stimulation of subsets of afferents, fluorescent microscopy, detailed reconstructions of the location of inhibitory synapses on the soma and dendrites of octopus cells, slice physiology, and computational modeling, they have been able to clarify the presence of functional inhibition and elucidate some of the features of the inhibitory inputs to octopus cells at a biophysical level. They also show through modeling that inhibition is predicted to both provide shunting of synaptic currents and to change the peak timing of dendritic EPSPs as they travel to the soma. Both of these effects are potentially critically important in integration in these fast, coincidence-detecting neurons, and the magnitudes of the effects could have physiological significance. Overall, this work extends thinking about the functional sensory processing roles of octopus cells beyond the pre-existing hypotheses that are focussed primarily on the coincidence detection of excitatory inputs.

The authors have addressed all of my prior concerns, including improving several aspects of the presentation. The modeling is better described, which is critical because it provides a foundation to help interpret some of the physiology and to propose specific functions.

---

## [Author Response]

The following is the authors’ response to the original reviews.

**eLife assessment**
This valuable work analyzes how specialized cells in the auditory cells, known as the octopus cells, can detect coincidences in their inputs at the submillisecond time scale. While previous work indicated that these cells receive no inhibitory inputs, the present study unambiguously demonstrates that these cells receive inhibitory glycinergic inputs. The physiologic impact of these inputs needs to be studied further. It remains incomplete at present but could be made solid by addressing caveats related to similar sizes of excitatory postsynaptic potentials and spikes in the octopus neurons.

We apologize for not explicitly describing our experimental methods and analyses procedures that ensure the discrimination between action potentials and EPSPs. This has been addressed in responses to reviewer comments and amended in the manuscript.

**Reviewer #1 (Public Review):**
Kreeger and colleagues have explored the balance of excitation and inhibition in the cochlear nucleus octopus cells of mice using morphological, electrophysiological, and computational methods. On the surface, the conclusion, that synaptic inhibition is present, does not seem like a leap. However, the octopus cells have been in the past portrayed as devoid of inhibition. This view was supported by the seeming lack of glycinergic fibers in the octopus cell area and the lack of apparent IPSPs. Here, Kreeger et al. used beautiful immunohistochemical and mouse genetic methods to quantify the inhibitory and excitatory boutons over the complete surface of individual octopus cells and further analyzed the proportions of the different subtypes of spiral ganglion cell inputs. I think the analysis stands as one of the most complete descriptions of any neuron, leaving little doubt about the presence of glycinergic boutons.Kreeger et al then examined inhibition physiologically, but here I felt that the study was incomplete. Specifically, no attempt was made to assess the actual, biological values of synaptic conductance for AMPAR and GlyR. Thus, we don't really know how potent the GlyR could be in mediating inhibition. Here are some numbered comments:(1) "EPSPs" were evoked either optogenetically or with electrical stimulation. The resulting depolarizations are interpreted to be EPSPs. However previous studies from Oertel show that octopus cells have tiny spikes, and distinguishing them from EPSPs is tricky. No mention is made here about how or whether that was done. Thus, the analysis of EPSP amplitude is ambiguous.

We agree that large EPSPs can be difficult to distinguish from an octopus cell’s short spikes during experiments. During analysis, we distinguished spikes from EPSPs by generating phase plots, which allow us to visualize the first derivative of the voltage trace on the y-axis and the value of the voltage on the x-axis at each moment in time. In the example shown below, four depolarizing events were electrically evoked in an octopus cell (panel A). The largest of these events (shown in orange in panels B-D) has an amplitude of ~9mV and could be a small spike. The first derivative of the voltage (panel C) reveals a bi-phasic response in the larger orange trace, where during the rising phase (mV/ms > 0) of the EPSP there is a second, sharper rising phase for the spike. Like more traditionally sized action potentials, phase plots for octopus cell spikes also reveal a sharp change in the rate of voltage change over time (Author response image 1 panel D, ✱) after the rising action of the EPSP begins to slow. EPSPs (shown in blue in panels B-D) lack the deflection in the phase plot. Not all cases were as unambiguous as this example. Therefore, our analysis only included subthreshold stimulation that unambiguously evoked EPSPs, not spikes. A brief description of this analysis has been added to the methods text (lines 625-627) and we have noted in the results section that both ChR2-evoked and electrically-evoked stimulation can produce small action potentials, which were excluded from analysis (lines 156-158).

(2) For this and later analysis, a voltage clamp of synaptic inputs would have been a simple alternative to avoid contaminating spikes or shunts by background or voltage-gated conductances. Yet only the current clamp was employed. I can understand that the authors might feel that the voltage clamp is 'flawed' because of the failure to clamp dendrites. But that may have been a good price to pay in this case. The authors should have at least justified their choice of method and detailed its caveats.

We agree that data collected using voltage-clamp would have eliminated the confound of short action potentials and avoided the influence of voltage-gated conductances. The large-diameter and comparatively simple dendritic trees of octopus cells make them good morphological candidates for reliable voltage clamp. However, as suggested, we were concerned that the abundance of channels open at the neuron’s resting potential would make it difficult to sufficiently clamp dendrites. Ultimately, given the low input resistances of octopus cells and the fast kinetics of excitatory inputs, we determined that bad voltage clamp conditions were likely to result in unclamped synaptic events with unpredicted distortions in kinetics and attenuation (To et al. 2022; PMID: 34480986; DOI: 10.1016/j.neuroscience.2021.08.024). We therefore chose to focus our efforts on current-clamp.

Beyond the limits of both current-clamp and voltage-clamp, we chose to leave all conductances that influence EPSP dendritic propagation intact because our model demonstrates that active Kv and leak conductances shape and attenuate synaptic inputs as they travel through the dendritic tree (Supp. Fig. 4F-G). The addition of voltage-clamp recordings would not impact the conclusions we make about EPSP summation at the soma. Future studies will need to focus on a dendrite-centric view of local excitatory and inhibitory summation. For dendrite-centric experiments, dendritic voltage-clamp recordings are well suited to answer that set of questions.

(3) The modeling raised several concerns. First, there is little presentation of assumptions, and of course, a model is entirely about its assumptions. For example, what excitatory conductance amplitudes were used? The same for inhibitory conductance? How were these values arrived at? The authors note that EPSGs and IPSGs had peaks at 0.3 and 3 ms. On what basis were these numbers obtained? The model's conclusions entirely depend on these values, and no measurements were made here that could have provided them. Parenthetical reference is made to Figure S5 where a range of values are tested, but with little explanation or justification.

We apologize for not providing this information. We used our octopus neuron model to fit both EPSP and IPSP parameters to match experimental data. We have expanded the methods to include final values for the conductances (lines 649-651), which were adjusted to match experimental values seen in current-clamp recordings. We have also expanded the results section to describe each of the parameters we tuned (lines 203-222). An example of these adjustments is illustrated in Fig. 4F where the magnitude of inhibitory potentials at different conductances (100nS and 1nS) was compared to experimental data over a range of octopus cell input resistance conditions. Kinetic parameters were determined by aligning modeled PSPs to the rise times and full width at half maximum (FWHM) measurements from experiments under control and Kv block conditions. The experimental data for EPSPs and IPSPs that was used to fit the model is shown in Author response image 2 below.

**Author response image 2. sa3fig2:** 

(4) In experiments that combined E and I stimulation, what exactly were time courses of the conductance changes, and how 'synchronous' were they, given the different methods to evoke them? (had the authors done voltage clamp they would know the answers).

We chose to focus data collection on voltage changes at the soma under physiological conditions to better understand how excitation and inhibition integrate at the somatic compartment. Our conclusions in the combined E and I stimulation experiments require the resting membrane properties of octopus cells to be intact to make physiologically-relevant conclusions. Our current-clamp data includes the critical impact of leak, Kv, and HCN conductances on this computation. Reliable voltage-clamp would necessitate the removal of the Kv and HCN conductances that shape PSP magnitude, shape, and speed. Because it was not necessary to measure the conductances and kinetics of specific channels, we chose to use current-clamp.

Evoked IPSPs and EPSPs had cell-to-cell variability in their latencies to onset. Somatically-recorded optically-evoked inhibition under pharmacological conditions that changed cable properties had onset latencies between 2.5 and 4.3ms; electrically-evoked excitation under control conditions had latencies between 0.8 and 1.4ms. To overcome cell-to-cell timing variabilities, we presented a shuffled set of stimulation pairings that had a 3ms range of timings with 200µs intervals. As the evoked excitation and inhibition become more ‘synchronous’, the impact on EPSP magnitude and timing is greatest. Data presented in this paper was for the stimulation pairings that evokes a maximal shift in EPSP timing. On average, this occurred when the optical stimulation began ~1.2ms before electrical stimulation. Stimulation pairing times ranged between a 0ms offset and a 1.8ms offset at the extremes. An example of the shuffled stimulation pairings is shown in Author response image 3 below, and we have included information about the shuffled stimulus in the methods (lines 627-630)

**Author response image 3. sa3fig3:** 

(5) Figure 4G is confusing to me. Its point, according to the text, is to show that changes in membrane properties induced by a block of Kv and HCN channels would not be expected to alter the amplitudes of EPSCs and IPSCs across the dendritic expanse. Now we are talking about currents (not shunting effects), and the presumption is that the blockers would alter the resting potential and thus the driving force for the currents. But what was the measured membrane potential change in the blockers? Surely that was documented. To me, the bigger concern (stated in the text) is whether the blockers altered exocytosis, and thus the increase in IPSP amplitude in blockers is due BOTH to loss of shunting and increase in presynaptic spike width. Added to this is that 4AP will reduce the spike threshold, thus allowing more ChR2-expressing axons to reach the threshold. Figure 4G does not address this point.

These are valuable points that motivated us to improve the clarity of this figure and the corresponding text. We discussed two separate points in this paragraph and were not clear. Our intention with Figure 4G was to address concerns that using pharmacological blockers changes driving forces and may confound the measured change in magnitude of postsynaptic potentials. Membrane potentials hyperpolarized by approximately 8-10 mV after application of blockers. We corrected for this effect by adding a holding current to depolarize the neuron to its baseline resting potential. Text in the results (lines 187-190) and figure legends have been changed to clarify these points.

We also removed any discussion of presynaptic effects from this portion of the text because our description was incomplete and we did not directly collect data related to these claims. We originally wrote, “While blocking Kv and HCN allowed us to reveal IPSPs at the soma, 4-AP increases the duration of the already unphysiological ChR2-evoked presynaptic action potential (Jackman et al., 2014; DOI: 10.1523/jneurosci.4694-13.2014), resulting in altered release probabilities and synaptic properties, amongst other caveats (Mathie et al., 1998; DOI: 10.1016/S0306-3623(97)00034-7)”. Ultimately, effects on exocytosis, presynaptic excitability, or release probability are only relevant for the experiments presented in Figure 4. Figure 4 serves as evidence that synaptic release of glycine elicits strychnine-sensitive inhibitory postsynaptic potentials in octopus cells. Concerns of presynaptic effects do not carry over to the data presented in Figure 5, as Kv and HCN were not blocked in these experiments. Therefore, we have removed this portion of the text.

(6) Figure 5F is striking as the key piece of biological data that shows that inhibition does reduce the amplitude of "EPSPs" in octopus cells. Given the other uncertainties mentioned, I wondered if it makes sense as an example of shunting inhibition. Specifically, what are the relative synaptic conductances, and would you predict a 25% reduction given the actual (not modeled) values?

We agree that both shunting and hyperpolarizing inhibition could play a role in the measured EPSP changes. Because we focused data collection on voltage changes at the soma under physiological conditions, we cannot calculate the relative synaptic conductances. Together, our experimental current-clamp results paired with estimates from the model provide compelling evidence for the change we observe in EPSPs. Regardless, the relative weights of the synaptic conductances is a very interesting question, but this information is not necessary to answer the questions posed in this study, namely the impact of dendritic inhibition on the arrival of EPSPs in the soma.

(7) Some of the supplemental figures, like 4 and 5, are hardly mentioned. Few will glean anything from them unless the authors direct attention to them and explain them better. In general, the readers would benefit from more complete explanations of what was done.

We apologize for not fully discussing these figures in the results text. We have fully expanded the results section to detail the experiments and results presented in the supplement (lines 203-238).

**Reviewer #2 (Public Review):**
Summary:Kreeger et.al provided mechanistic evidence for flexible coincidence detection of auditory nerve synaptic inputs by octopus cells in the mouse cochlear nucleus. The octopus cells are specialized neurons that can fire repetitively at very high rates (> 800 Hz in vivo), yield responses dominated by the onset of sound for simple stimuli, and integrate auditory nerve inputs over a wide frequency span. Previously, it was thought that octopus cells received little inhibitory input, and their integration of auditory input depended principally on temporally precise coincidence detection of excitatory auditory nerve inputs, coupled with a low input resistance established by high levels of expression of certain potassium channels and hyperpolarization-activated channels.In this study, the authors used a combination of numerous genetic mouse models to characterize synaptic inputs and enable optogenetic stimulation of subsets of afferents, fluorescent microscopy, detailed reconstructions of the location of inhibitory synapses on the soma and dendrites of octopus cells, and computational modeling, to explore the importance of inhibitory inputs to the cells. They determined through assessment of excitatory and inhibitory synaptic densities that spiral ganglion neuron synapses are densest on the soma and proximal dendrite, while glycinergic inhibitory synaptic density is greater on the dendrites compared to the soma of octopus cells. Using different genetic lines, the authors further elucidated that the majority of excitatory synapses on the octopus cells are from type 1a spiral ganglion neurons, which have low response thresholds and high rates of spontaneous activity. In the second half of the paper, the authors employed electrophysiology to uncover the physiological response of octopus cells to excitatory and inhibitory inputs. Using a combination of pharmacological blockers in vitro cellular and computational modeling, the authors conclude that glycine in fact evokes IPSPs in octopus cells; these IPSPs are largely shunted by the high membrane conductance of the cells under normal conditions and thus were not clearly evident in prior studies. Pharmacological experiments point towards a specific glycine receptor subunit composition. Lastly, Kreeger et. al demonstrated with in vitro recordings and computational modeling that octopus cell inhibition modulates the amplitude and timing of dendritic spiral ganglion inputs to octopus cells, allowing for flexible coincidence detection.Strengths:The work combines a number of approaches and complementary observations to characterize the spatial patterns of excitatory and inhibitory synaptic input, and the type of auditory nerve input to the octopus cells. The combination of multiple mouse lines enables a better understanding of and helps to define, the pattern of synaptic convergence onto these cells. The electrophysiology provides excellent functional evidence for the presence of the inhibitory inputs, and the modeling helps to interpret the likely functional role of inhibition. The work is technically well done and adds an interesting dimension related to the processing of sound by these neurons. The paper is overall well written, the experimental tests are well-motivated and easy to follow. The discussion is reasonable and touches on both the potential implications of the work as well as some caveats.Weaknesses:While the conclusions presented by the authors are solid, a prominent question remains regarding the source of the glycinergic input onto octopus cells. In the discussion, the authors claim that there is no evidence for D-stellate, L-stellate, and tuberculoventral cell (all local inhibitory neurons of the ventral and dorsal cochlear nucleus) connections to octopus cells, and cite the relevant literature. An experimental approach will be necessary to properly rule out (or rule in) these cell types and others that may arise from other auditory brainstem nuclei. Understanding which cells provide the inhibitory input will be an essential step in clarifying its roles in the processing of sound by octopus cells.

We are glad that the reviewer agrees with the conclusions we have made and is interested in learning more about how these findings impact sound processing. We agree that defining the source of inhibition will dramatically shape our understanding of the computation octopus cells are making. However, this is not an easy task, given the small size of the octopus cell area, and will involve considerable additional work. Since the overall findings do not depend on knowing the source of inhibition, we have instead re-written the discussion to clarify the lack of evidence for intrinsic inhibitory inputs to octopus cells, in addition to presenting likely candidates. As genetic profiles of cochlear nucleus and other auditory brainstem neurons become available, we intend to make and utilize genetic mouse models to answer questions like this.

The authors showed that type 1a SGNs are the most abundant inputs to octopus cells via microscopy. However, in Figure 3 they compare optical stimulation of all classes of ANFs, then compare this against stimulation of type 1b/c ANFs. While a difference in the paired-pulse ratio (and therefore, likely release probability) can be inferred by the difference between Foxg1-ChR2 and Ntng1-ChR2, it would have been preferable to have specific data with selective stimulation of type 1a neurons.

We agree that complete genetic access to only the Ia population would have been the preferable approach, but we did not have an appropriate line when beginning these experiments. Because our results did not suggest a meaningful difference between the populations, we did not pursue further investigation once a line was available.

**Recommendations for the authors:**

**Reviewer #1 (Recommendations For The Authors):**
Besides the points mentioned in the main review:Minor(1) I really like the graphics and the immunohistological presentation.(2) Lines 316-319 say that octopus cells lack things like back-propagating spikes and dendritic Ca spikes. How do you know this?

This statement was intended to be a summary of suggestions from the literature and lacked references and context as written. We have rewritten this section and clarified that our hypothesis was formed from data found in the literature (lines 334-337).

(3) Spectrograms of Figure 6A...where were these data obtained?

We recorded and visualized human-generated rhythmic tapping and high-frequency squeaking sounds using Audacity. The visualizations of rhythmic tapping and imitated vocalizations are meant to show two different types of multi-frequency stimuli we hypothesize would result in somatic summation within an octopus cell’s spike integration window, despite differences in timing. We rewrote the figure legend to explain more clearly what is shown and how it relates to the model in Figure 6.

(4) 'on-path' and 'off-path' seem like jargon that may not be clear to the average reader.

Thank you for pointing out our use of unapproachable jargon. We have replaced the term from the figure with “proximal” and “distal” inhibition. In the main text, we now describe on-path and off-path together as the effect of location of dendritic inhibition on somatically recorded EPSPs.

(5) The paper could benefit from a table of modeled values.

We have added specific details about the modelling in the text and clarified which modeled values were referenced from previous computational models and which were tuned to fit experimental data. Since most values were taken from a referenced publication, we did not add a table and instead point readers towards that source.

(6) Figure S4A-C what currents were delivered to the modeled cells?

The model cells were injected with a -0.8 nA DC current for 300 ms in current clamp mode. This information has been added to the figure legend.

(7) In that figure "scaling factors" scale exactly which channels?

Scaling factor is used to scale low-voltage activated K^+^ (ḡ_KLT_), high threshold K^+^ (ḡ_KHT_), fast transient K^+^ (ḡ_KA_), hyperpolarization-activated cyclic nucleotide-gated HCN (ḡ_h_) but not fast Na^+^ (ḡ_Na_) and leak K^+^ (ḡ_leak_). This information has been added to the text (lines 205-208 and 646-653).

(8) In performing and modeling Kv/HCN block, do you know how complete the level of the block is?

Since we cannot assess how complete the level of block is, we have changed the language in the text to clarify that we are reducing Kv and HCN channel conductance to the degree needed to increase resistance of the neuron (line 185).

(9) More on this Figure S4. It is hardly referred to in the text except to say that it supports that blocking the Kv/HCN channels will enhance the IPSP. Given how large the figure is, can you offer more of a conclusion than that? Also, in the synaptic model in that figure, the IPSCs are presumably happening in current-clamp conditions, and the reduction in amplitude of the IPSC (as opposed to the increase in IPSP) is due to hyperpolarization. Can you simply state that so readers can track what this figure is showing? Other similar things: what is a transfer impedance? How is it measured? What do we take from the analysis?

We have elaborated on our description of both Supp. Fig. 4 and Supp. Fig. 5 in the results section of the text (lines 203-238).

(10) Figure S5 also needs a better explanation. E.g., in C-D, what does 'average' mean? The gray is an SD of this average? You modeled a range of values...but which ones are physiological? To me, this is a key point.

We have elaborated on our description of both Supp. Fig. 4 and Supp. Fig. 5 in the results section of the text (lines 203-238).

**Reviewer #2 (Recommendations For The Authors):**
General:The images and 3-D reconstructions are visually stunning, but they are not colorblind-friendly and in some cases, hard to distinguish. This shows up particularly in the green and blue colors used in Figure 1. Also, better representative images could be used for Figure 1B.

Thank you for pointing out that blue and green were difficult to distinguish in Figure 1H. We have outlined the green inhibitory puncta in this image to make them more distinguishable. We have also increased the resolution of the image in Figure 1B for better clarity. All other colors are selected from Wong, 2011 (PMID: 21850730; DOI: https://doi.org/10.1038/nmeth.1618).

Supplemental Figure 1D: The low-power view is good to have, but the CN is too small and the image appears a bit noisy. An inset showing the CN on a larger scale (higher resolution image?) would be more convincing. In this image, I see what appear to be cells in the DCN labeled, which calls into question the purity of the source of optogenetic synaptic activation. It is also difficult to tell whether there are other cells labeled in the VCN. Such inputs would still be minor, but it would be good to be very clear about the expression pattern.

To offer more information about the activity of the *Ntng1*^Cre^ line in other regions of the auditory system, we increased the resolution of the image included in Supp. Fig. 1D and have also included an additional image (Supp. Fig. 1E) of a coronal section of the cochlear nucleus complex with *Ntng1*-tdT labelling. This image provides additional context for the cells labeled in the DCN. The text in the figure legend has been changed to clarify that some cells in the DCN were labeled (lines 118-120).

We agree that in the *Ntng1*^Cre^ experiments, there is the possibility of minor contamination from excitatory cells that express ChR2 outside of the spiral ganglion. This is also true for our *Foxg1*^Cre^ and *Foxg1*^Flp^ experiments, because these lines label cortical cells in addition to cochlear cells. However, we do not observe direct descending inputs from the cortex into the PVCN, making contamination from other *Foxg1*^Cre^-positive neurons unlikely. While non-cochlear inputs from the *Ntng1*^Cre^ line are possible, evidence from both lines gives us confidence that we are not capturing inputs to octopus cells outside the cochlea. Central axons from Type I spiral ganglion neurons have VGLUT1+ synaptic terminals. When comparing the overlap between VGLUT1+ terminals and *Foxg1*-tdT labelling, we see full coverage. That is, all VGLUT+ terminals on octopus cells are co-labelled by *Foxg1*^Cre^-mediated expression of tdTomato. An example image is shown below. Here, an octopus cell soma is labeled with blue fluorescent Nissl stain and inputs to the cochlear nucleus complex are labeled with *Foxg1*^Cre^-dependent tdTomato (*Foxg1*-tdT; magenta). We have also immunolabeled for VGLUT1 puncta in green. This eliminates the possibility that VGLUT+ cells from outside the cochlea and cortex are sources of excitation to octopus cells.

**Author response image 4. sa3fig4:** 

Further, we have looked at expression of *Ntng1*-tdT and *Foxg1*-EYFP together in the octopus cell area. An example image is shown below. All *Ntng1*-tdT+ fibers (magenta) are also *Foxg1*-EYFP+ (green), suggesting that all *Ntng1*^Cre^-targeted inputs to octopus cells are a part of the *Foxg1*^Cre^-targeted input population, which are very likely to only be from the cochlea. We have expanded the results section to include information about the overlap in expression driven by the *Ntng1*^Cre^ and *Foxg1*^Flp^ lines.

**Author response image 5. sa3fig5:** 

Supplemental Figure 2 G: These are a bit hard to read. Perhaps use a different image, or provide a reference outline drawing telling us what is what.

We have used a different image with a Thy1-YFP labeled octopus cell for clarity.

In some places, the term "SGN" is used when referencing the axons and terminals within the CN, and without some context, this was occasionally confusing (SGN would seem to refer to the cell bodies). In some places in the text, it may be preferable to separate SGN, auditory nerve fibers (ANFs), and terminals, as entities for clarity.

In order to make the study accessible to a broad neuroscience audience, we refer to the neurons of the spiral ganglion and their central axon projections using one name. We understand why, for those well acquainted with the auditory periphery, condensing terminology may feel awkward. However, for those readers unfamiliar with the anatomy of the cochlea and auditory nerve, we feel that the use of “SGN central axon” makes it clear that the “auditory nerve fibers” come from neurons in the spiral ganglion. This is clarified in the first paragraph of the introduction (lines 29-31) and in the methods (line 533).

Specific: Numbers refer to the line numbers on the manuscript.L29-31: Cochlear nucleus neurons are more general in their responses than this sentence indicates. While we can all agree that they are specialized to carry (or improve upon) the representation of these specific features of sound, they also respond more generally to sounds that might not have specific information in any of these domains. They are not silos of neural computation, and their outputs become mixed and "re-represented" well before they reach the auditory cortex. Octopus cells are no exception to this. I suggest striking most of the first paragraph, and instead using the first sentence to lead into the second paragraph, and putting the last sentence (of the current first paragraph) at the end of the second (now first) paragraph.

We agree with this assessment and have made major changes to the introduction in line with these suggestions.

L33-46: A number of points in this paragraph need references (exp. line 41).

We agree and have added references accordingly.

L43: Not sure what is meant by "fire at the onset of the sound, breaking it up into its frequency components"?

We changed this text as part of a major reworking of the introduction.

L47-66: Again more citations are needed (at the end of sentence at line 55, probably moving some of the citations from the next sentence up).

We agree and have added references accordingly.

L51: The consistent orientation of octopus cell dendrites across the ANFs has been claimed in the literature (as mentioned here), but there are some (perhaps problematic - plane of sectioning?) counterexamples from the older Golgi-stained images, and even amongst intracellularly stained cells (for example see Reccio-Spinoza and Rhode, 2020). This is important with regards to the broader hypothesis regarding traveling-wave compensation (e.g., McGinley et al; but also many others); if the cells are not all in the appropriate orientation then such compensation may be problematic. Likewise, the data from Lu et al., 2022, points towards a range of sensitivity to frequency-swept stimuli, some of which work in opposition to the traveling wave compensation hypothesis. It would seem that with the Thy1 mice, you have an opportunity to clarify the orientation. Figures 1A and 2A show a consistent dendritic orientation, assuming that these drawings are reconstructions of the cells as they were actually oriented in the tissue. Can you either comment on this or provide clearer evidence?

We are happy to offer more information about the appearances of octopus cells in our preparations. In our hands, sparsely labeled octopus cells in Thy1-YFP-H mice show consistent dendritic orientation when visualized in a 15 degree parasaggital plane, with the most diversity apparent in cells with somas located more dorsally in the octopus cell area. We hypothesize that this is due to the limited area through which the central projections of spiral ganglion neurons (i.e. ANFs) must pass through before they enter the dorsal cochlear nucleus and continue their tonotopic organization in that area.

A caveat to studies without physiological or genetic identification of octopus cells is the assumption that all neurons in the octopus cell area are octopus cells. We find, especially along the borders of the octopus cell area, that stellate cells can be seen amongst octopus cells. Because stellate cell dendrites are not oriented like octopus cell dendrites, any stellate cells misidentified as octopus cells would appear to have poorly-oriented dendrites. This may explain why some studies report this finding. In addition, it can be difficult to assess tonotopic organization because of the 3D trajectory of tightly bundled axons, which is not capturable by a single section plane. Although a parasaggital plane of sectioning captures the tonotopic axis in one part of the octopus cell area, that same plane may be perpendicular at the opposing end.

L67: canonical -> exceptional.

Thank you for the suggestion. We have made this change in the introduction.

L127: This paragraph was confusing on first reading. I don't think Supplemental Figure 1D shows the restricted pattern of expression very clearly. The "restricted to SGNs" might be better as "restricted to auditory nerve fibers" (except in the DCN, where there seem to be some scattered small cells?). A higher magnification image of the CN, but lower magnification than in panel E, would be helpful here.

To avoid confusion, we have re-written this paragraph (lines 117-127) and included a higher magnification image of the CN in a revised Supp. Fig. 1.

L168: Here, perhaps say ANFs instead of SGNs.

As above, we have decided to describe ANFs as SGN central axons to make the anatomy more accessible to people unfamiliar with cochlear anatomy.

L201-204: The IPSPs are surprisingly slow (Figures 5B, C), especially given the speed of the EPSPs/EPSCs in these cells. This is reminiscent of the asymmetry between EPSC and IPSC kinetics in bushy cells (Xie and Manis, 2014). The kinetics used in the model (3 ms; mentioned on line 624) however seem a bit arbitrary and no data is provided for the selection of that value. Were there any direct measurements of the IPSC kinetics (all of the traces in the paper are in the current clamp) that were used to justify this value?

The kinetics of the somatically-recorded IPSPs are subject to the effects of our pharmacological manipulations. EPSPs measured at the soma under control conditions are small amplitude and rapid. With pharmacological reduction of HCN and Kv channels, EPSPs are larger and slower (please see figure in response to a similar question posed by Reviewer #1). We expect that this change also occurs with the IPSP kinetics under pharmacological conditions. Our justification of kinetics has been expanded and justified in the methods section (lines 641-661).

L594: Technically, this is a -11 mV junction potential, but thanks for including the information.

We have corrected this in the text (line 618). Thank you for the close reading of all experimental and methodological details.

L595: The estimated power of the LED illumination at the focal plane should be measured and indicated here.

We measured the power of the LED illumination at the focal plane using a PM100D Compact Power and Energy Meter Console (Thorlabs), a S120C Photodiode Power Sensor (Thorlabs), and a 1000µm diameter Circular Precision Pinhole (Thorlabs). Light intensity at the focal plane ranged between 1.9 and 4.1mW/mm^2^, corresponding to 6% and 10% intensity on the Colibri5 system. We have reported these measurements in the results section (Lines 621-622).

L609: One concern about the model is that the integration time of 25 microseconds is rather close to the relative shifts in latency. While I doubt it will make a difference (except in the number), it may be worth verifying (spot checks, at least) that running the model with a 5 or 10-microsecond step yields a similar pattern of latency shifts (e.g., Supplementary Figure 5, Figure 5).

Also, it is not clear what temperature the model was executed at (I would presume 35C); this needs to be given, and channel Q10's listed.

We realize that additional information is needed to fully understand the model and have added this to the results and the methods. The synaptic mechanism (.mod) files were obtained from Manis and Campagnola, 2018. Q10 (3) and temperature (22°C) were also matched to parameters from Manis and Campagnola, 2018. Because temperature is a critical factor for channel kinetics, we verified that our primary results remain consistent under conditions using a temperature of 35°C and a time step of 5µs, depicted below. Panel A illustrates the increase in IPSP as a function of glycine conductance under Kv+HCN block conditions at 35°C. As at 22°C, an increase in IPSP magnitude is absent in the control condition at 35°C. Panels B and C provide a direct comparison between the initial (i.e. 22°C) and suggested (i.e. 35°C) simulation conditions. Again we found that temperature does not have a major impact on the amplitude of IPSPs. Thus, results at 35°C do not change the conclusions we make from the model.

**Author response image 6. sa3fig6:** 

The nominal conductance densities should at least be provided in a table (supplemental, in addition to including them in the deposited code). The method for "optimization" of the conductance densities to match the experimental recordings needs to be described; the parameter space can be quite large in a model such as this. The McGinley reference needs a number.

We added a more thorough description of modeling parameters and justification of choices in the methods section of the text (lines 641-661). We have also added a reference number to the McGinley 2012 reference in the text.

I think this is required by the journal:The model code, test results, and simulation results should be deposited in a public resource (Github would be preferable, but dryad, Zenodo, or Figshare could work), and the URL/doi for the resource provided in the manuscript. This includes the morphology swc/hoc file. The code should be in a form, and with a description, that readily allows an interested party with appropriate skills to download it and run it to generate the figures.

We will upload the code and all associated simulation files to the ModelDB repository upon publication.